# Rationally designed mineralization for selective recovery of the rare earth elements

Takaaki Hatanaka[1], Akimasa Matsugami[2], Takamasa Nonaka[1], Hideki Takagi[1], Fumiaki Hayashi[2], Takao Tani[1] & Nobuhiro Ishida[1]

The increasing demand for rare earth (RE) elements in advanced materials for permanent magnets, rechargeable batteries, catalysts and lamp phosphors necessitates environmentally friendly approaches for their recovery and separation. Here, we propose a mineralization concept for direct extraction of RE ions with Lamp (lanthanide ion mineralization peptide). In aqueous solution containing various metal ions, Lamp promotes the generation of RE hydroxide species with which it binds to form hydrophobic complexes that accumulate spontaneously as insoluble precipitates, even under physiological conditions (pH ∼6.0). This concept for stabilization of an insoluble lanthanide hydroxide complex with an artificial peptide also works in combination with stable scaffolds like synthetic macromolecules and proteins. Our strategy opens the possibility for selective separation of target metal elements from seawater and industrial wastewater under mild conditions without additional energy input.

[1] Toyota Central R&D Labs., Inc., 41-1, Nagakute, Aichi 480-1192, Japan. [2] NMR Facility Support Unit, Division of Structural and Synthetic Biology, RIKEN Center for Life Science Technology, 1-7-22 Suehiro-cho, Tsurumi-ku, Yokohama, Kanagawa 230-0045, Japan. Correspondence and requests for materials should be addressed to N.I. (email: e1168@mosk.tytlabs.co.jp).

Lanthanide elements (Ln) are a class of rare earth (RE) species consisting of 15 elements with extremely similar character. The distinctive electronic features provide special magnetic and optical properties that allow Ln to be applied in various advanced equipment, including hybrid and electric cars, wind turbines, light emitting diode lamps and mobile phones[1,2]. The global supply of Ln is at risk from the ever-increasing demand. For sustainable utilization, more environmentally and economically friendly separation/recovery techniques than those currently in use, such as solvent extraction or ion chromatography, are required to directly target scarce Ln distributed in nature and urban mines[2,3]. Several studies recently reported the extraction and separation of Ln from transition metals or other Ln species. For instance, a hollow fibre membrane system combined with neutral extractants, such as tetraoctyldiglycolamide, was shown to selectively recover Ln species[4,5]. A metal–organic framework based on homochiral camphorate derivatives, which exhibits ionic radii selective crystallization, was applied to separate light and heavy $Ln^{3+}$ (ref. 6). Moreover, a tripod nitroxide ligand successfully separated $Dy^{3+}$ and $Nd^{3+}$ based on size-sensitive dimerization[7]. In principle, all of these attractive approaches exploit electrostatic interactions with $Ln^{3+}$ and exhibit controllable accumulation ability. However, these require environmentally unfavourable artificial treatments, such as long incubation times, high temperatures, low pH and the use of organic solvents. Thus, other techniques are required for RE separation/recovery under physiological conditions with low energy input.

Biomineralization is a unique phenomenon that proceeds spontaneously under low-energy conditions in living organisms to form organic/inorganic composite materials, such as teeth, bone and pearls[8–11]. These sophisticated systems are controlled by biomolecule functionalities that selectively recognize target ions (or molecules) and self-organization[12–15], which has fuelled interest in biological material design[16]. One successful effort is the use of artificial peptides, which recognize transition metals and metal compounds (Au, Ag, Pt, $TiO_2$, CuO, ZnO and so on), for nanostructure construction[17–22]. For example, a ZnS-binding peptide, able to control ZnS nanocrystal size, enabled the formation of ZnS quantum dots by combining with the M13 phage display system[23,24]. Another peptide targeting Pt could control Pt nanocrystal shapes, including square, pyramid and ninja star structures[25,26]. These metal-recognizing peptides show selective binding of the crystal face and can control the size, shape and composition of crystal growth thermodynamically. However, most of these interesting peptides only work with the assistance of reducing agents, which are necessary to initiate mineralization, and none can directly mineralize metal ions under conditions where the metal is stable as an ion. Consequently, metal-recognizing peptides have not been applied for direct extraction of Ln species because of the high stability of their ionic state.

Herein, we propose an artificial biomineralization concept based on a specific peptide. Considering the chemical equilibrium of Ln species in aqueous solution, the peptides were screened with Ln-hydroxide (hydro-oxide) using the T7 phage display technique. The identified peptides bound tightly to Ln-hydroxide and precipitated Ln minerals with low energy, even under conditions where the ionic form of the metal is stable. We also clarified the mineralization mechanism and the peptide function on stable scaffolds like proteins and polymer resin. The biomineralization-based system reported here provides an alternative approach for RE recovery.

## Results

**Design concept for lanthanide ion mineralization.** Our concept uses an artificial peptide that stabilizes an insoluble lanthanide hydroxide complex (Fig. 1). In aqueous solution, metal ions commonly form complexes with $H_2O$ molecules. The electrophilicity of metal ions causes a hydroxylation/dehydroxylation equilibrium, and the equilibrium constant of this reaction, represented as Log $K$ ($K = [Ln(OH)]^{2+}/[Ln^{3+}][(OH)^-]$), is different for each metal (Supplementary Fig. 1). Accordingly, the equilibrium shifts from $Ln^{3+}$ to Ln monohydroxide $[Ln(OH)]^{2+}$ at pH 8.2–9.3 (Log $K$ = 4.7–5.8), which is the rate limiting step of trihydroxide generation, for which Log $K$ is 20.7–26.1 (ref. 27). We expected that generation, growth and precipitation of Ln hydroxides would be promoted by molecules that stabilize the Ln-OH interaction and prevent dehydroxylation, even at slightly acidic to neutral conditions. Thus, we aimed to obtain a peptide that tightly binds Ln hydroxide.

To effectively isolate such a peptide, we selected a cyclic peptide platform comprising 9–12 random sequences between two Cys residues, which form an intra-disulfide bond. The use of a cyclic peptide has been demonstrated to reduce unfavourable changes in conformational entropy for target recognition[28–31]. Moreover, referring to the sequences of metal-binding peptides, which contain higher numbers of hydrophilic amino acids than hydrophobic amino acids[18,21], we selected the NNK codon, which has a tendency in include 56.25% hydrophilic amino acids, to randomize the peptide sequence[32]. The peptide libraries were constructed on the T7 phage display system, which displays an average of 5–15 copies of the peptide on the phage surface. This multiple display system provides rebinding and the avidity effect, which makes the phage-binding tighter, even though the binding affinity is not high[33,34]. The diversity of the constructed peptide libraries was estimated as $1.56 \times 10^6 - 3.76 \times 10^7$ (Supplementary Table 1), which covers only a part of the theoretical diversity; $5.120 \times 10^{11} - 4.096 \times 10^{15}$ ($20^9 - 20^{12}$). However, owing to the appropriate designs, the constructed library was expected to isolate the target binder, even though it contained limited diversity. For peptide screening, hydroxylated $Dy_2O_3$ and

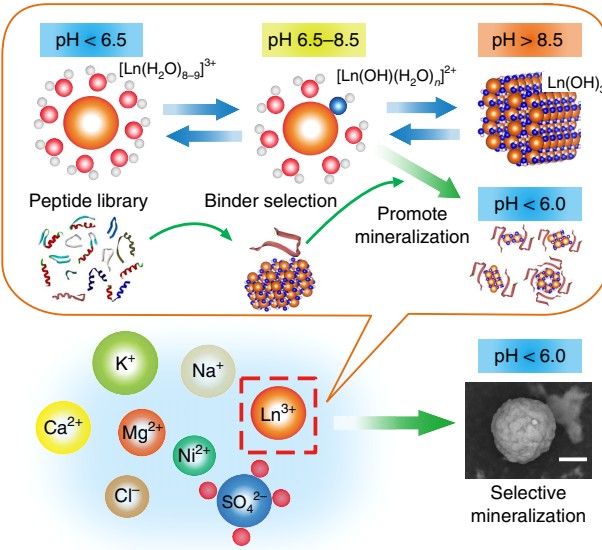

**Figure 1 | Schematic illustration of peptide-based selective mineralization.** $Ln^{3+}$ molecules (orange spheres) are coordinated by water molecules (red and grey spheres) in aqueous solution, which change morphology depending on pH. We aimed to select a peptide that recognizes $Ln(OH)_3$ (blue and grey spheres). The specific peptide should generate and precipitate Ln hydroxides even under slightly acidic conditions. Scale bar, 10 μm.

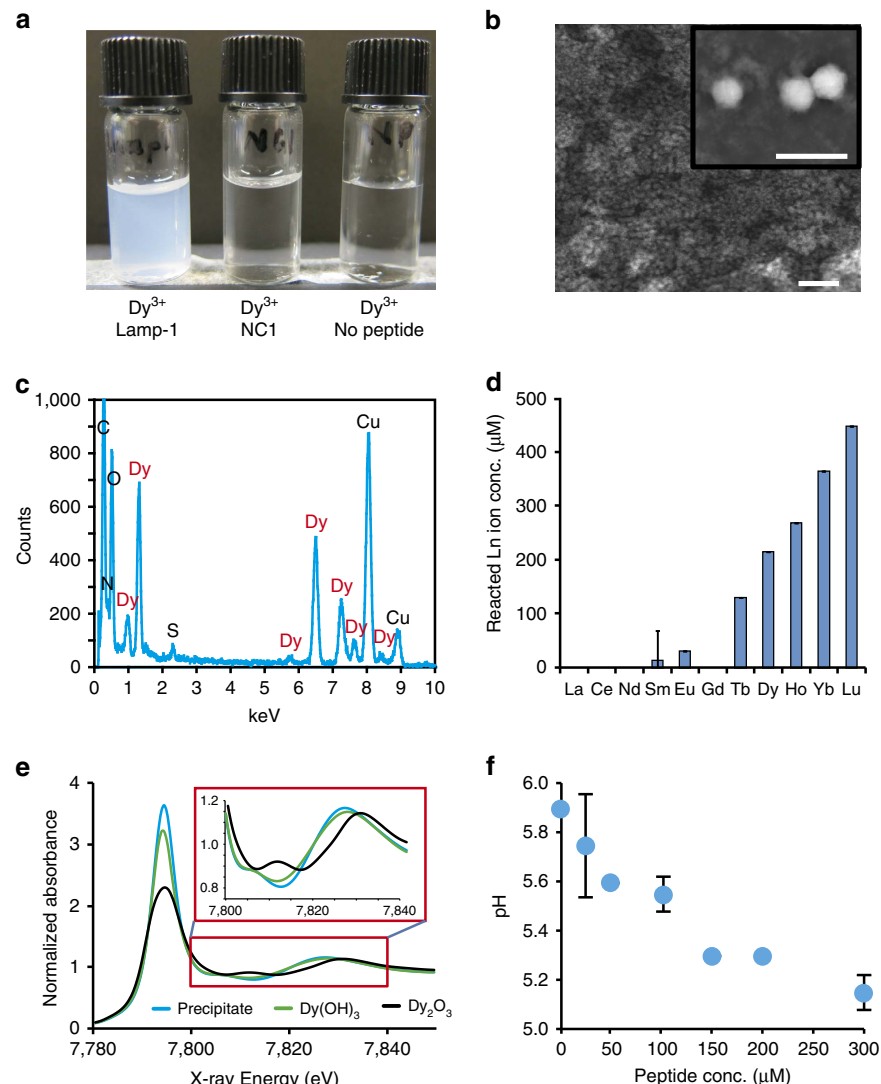

**Figure 2 | Generation of Dy hydroxide with Lamp-1 in neutral conditions.** (**a**) Optical image of mineralization induced by Lamp-1. NC1 is a control peptide that has little consensus with Lamp-1. (**b**) TEM (large panel) and SEM (small panel) images of the generated precipitate. Scale bars, 20 nm and 10 µm, respectively. (**c**) EDX analysis of the precipitate. The high copper signals result from the TEM grid. (**d**) The reactivity of Lamp-1 with a series of $Ln^{3+}$. The precipitated $Ln^{3+}$ concentration was analysed by ICP-OES ($n = 3$). Error bars denote s.d. (**e**) Normalized Dy $L_3$-edge X-ray absorption near edge structure spectrum of the precipitate (blue). The inset represents an expanded view of the red square. $Dy(OH)_3$ (green) and $Dy_2O_3$ (black) were used as controls. (**f**) Solution pH at different Lamp-1 concentrations ($n = 4$). Lamp-1 and $Dy(NO_3)_3$ were each dissolved in 0.1 mM MES buffer and the pH was adjusted to 5.9–6.0. The pH value (vertical axis) was measured after mixing these two solutions. Error bars denote s.d.

$Nd_2O_3$ nanoparticles (particle size < 100 nm), confirmed to have hydroxylated surfaces by Fourier transform infrared (FT-IR) spectroscopy, were used as the target (Supplementary Fig. 2). The surface of the Ln oxide nanoparticle is hydroxylated easily, though the degree of hydroxylation depends on the particle size[35]. By sequencing analysis after five rounds of bio-panning, three individual peptide were identified at high frequency from the randomly picked up phage clones (Supplementary Table 1). Although it was difficult to identify the consensus motif, these peptides contained aspartic acid (Asp) and no basic amino acids. In addition, the three phages showed higher binding ability than the other clones toward each hydroxylated $Ln_2O_3$ nanoparticle (Supplementary Fig. 3). These peptides were then synthesized by Fmoc chemistry (Supplementary Table 2) and their ability to bind the nanoparticles was analysed. Compared with two control peptides (LBT3 and RE-1) previously reported to recognize $Ln^{3+}$ and $Ln_2O_3$ (refs 36,37), our selected peptides showed a binding affinity that was a maximum of 242-fold higher for hydroxylated

$Ln_2O_3$ (Supplementary Table 3). With these characteristics, these three kinds of sequences were selected as candidates for lanthanide ion mineralization peptides (Lamp -1: SCLWGDVSELDFLCS, -2: SCLYPSWSDYAFCS and -3: SCPVWFSDVGDFMVCS).

**Peptide-based direct mineralization**. Adding synthetic Lamp-1 to a solution containing $Dy(NO_3)_3$ gave a cloudy mixture with a rapid increase in turbidity at pH 6.0 (Fig. 2a, Supplementary Fig. 4 and Supplementary Movie 1). This phenomenon was also observed for the chloride and acetate salts in various buffers, and precipitate formation increased with incubation time (Supplementary Fig. 5). Similar phenomena were also confirmed for Lamp-2 and -3, while control peptides (LBT3, RE-1 and NC1) did not show any visible changes (Supplementary Fig. 6). Scanning electron microscope (SEM) and transmission electron microscope (TEM) analyses showed that the precipitate consists

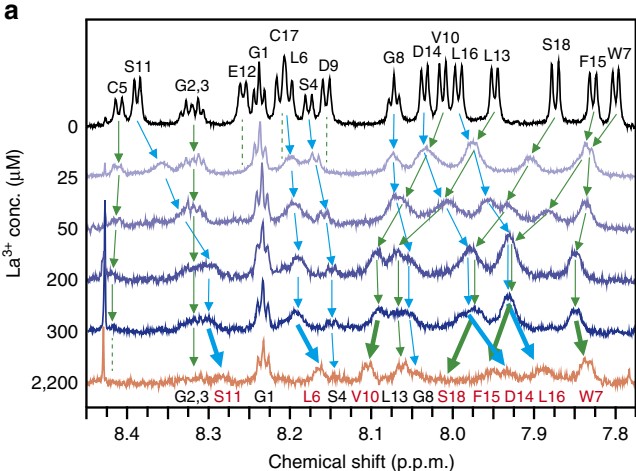

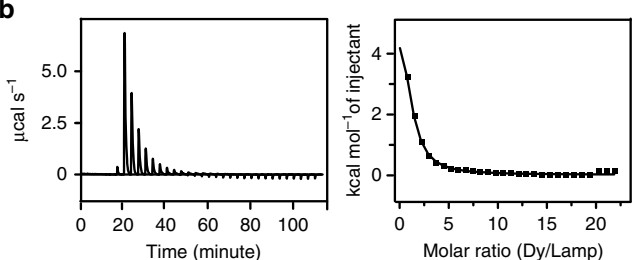

**Figure 3 | Detailed analysis of the mineralization phenomenon.** (**a**) $^1H$ NMR spectrum of Lamp-1. Chemical shift changes in the amide proton region of Lamp-1 with addition of $La^{3+}$. The dotted lines show peaks that disappear during the titration. The assignments in red and thick arrows indicate residues largely shifted after titration to 2,200 µM $La^{3+}$. (**b**) Thermodynamic analysis of the reaction between Lamp-1 and $Dy^{3+}$. The left panel shows a calorimetric titration profile. The right panel shows a least squares fit of the data for the heat absorbed/mol of titrant versus the ratio total $Dy^{3+}$ concentration to total Lamp-1 concentration. After the ITC measurements, a visible precipitate was observed in the solution that contained 90 µM of Lamp-1 and 2 mM of $Dy^{3+}$.

of subnanoscale structures that accumulate to form clusters (Fig. 2b). Furthermore, energy dispersive X-ray (EDX) spectroscopy and reverse phase chromatography indicated that the precipitate contains Dy and Lamp (Fig. 2c and Supplementary Fig. 7). As Ln species exhibit extremely similar properties, we evaluated the reaction selectivity against 11 typical $Ln^{3+}$ using inductively coupled plasma optical emission spectrometry (ICP-OES). Clearly, heavy Ln species were selectively precipitated over light Ln species (Fig. 2d), consistent with this, the consumption percentage of Lamp-1 for La and Lu increased from 5.5 to 80%, respectively (Supplementary Fig. 8). The stoichiometry also increased as the atomic number increased from about 1.4 (Tb) to 1.9 (Lu). This reaction tendency correlates highly with the stability constants of the Ln hydroxides. On the other hand, Lamp-2 and -3 showed non-selective reaction and precipitated La, Dy and Lu equally. The reaction stoichiometry of these peptides, which was calculated to be ~0.5, was also different from that of Lamp-1.

To confirm the precipitate contains a compound with $Ln(OH)_3$-like structure, we performed X-ray absorption fine structure (XAFS) measurements. The X-ray absorption near edge structure spectrum of the precipitate showed high similarity to that of $Dy(OH)_3$, but not with that of $Dy_2O_3$ (Fig. 2e). Extended XAFS analysis provided a similar result with the distance from Dy to oxygen in the first coordination sphere almost equal to that in

$Dy(OH)_3$, whereas no obvious peak was observed in the second coordination sphere, indicating that the precipitate forms an amorphous structure (Supplementary Fig. 9). Almost identical results were observed for Lamp-2 and -3. Furthermore, the pH of the solution containing $Ln^{3+}$ decreased with peptide addition (Fig. 2f and Supplementary Fig. 10), in agreement with the Ln hydroxylation reaction: $Ln^{n+} + nH_2O \rightleftharpoons Ln(OH)_n + nH^+$. These findings demonstrate that isolated peptides can precipitate Ln hydroxide directly from $Ln^{3+}$ under mild conditions, supporting our concept that an insoluble lanthanide hydroxide complex can be stabilized with an artificial peptide.

**Mechanism of peptide-based mineralization.** We closely examined the Lamp–$Ln^{3+}$ interaction using nuclear magnetic resonance (NMR) analysis to evaluate the reaction profiles. Titration with $Dy^{3+}$ perturbed the chemical shifts of Lamp-1 because of its high paramagnetic property[38]. The Asp9 backbone and acidic side chains were especially sensitive, with signals significantly diminished, even at the lowest $Dy^{3+}$ concentration. This result clearly indicates that the coordination of $Dy^{3+}$ by the carboxylate occurs first during reaction with Lamp, which is similar to the features observed for chelation with $Ln^{3+}$ (refs 36,39,40). Increasing the concentration of $Dy^{3+}$ to 2 µM resulted in the disappearance of half the NH resonances, mainly those of C-terminal residues (Supplementary Figs 11 and 12). Similarly, titration with $La^{3+}$-induced chemical shift perturbations, especially for acidic residues Asp9 and Glu12 (Fig. 3a). Moreover, the diamagnetic character of $La^{3+}$ enabled observation of two distinct titration phases. In the first phase, most of the resonance peaks shifted and broadened during titration to 200 µM $La^{3+}$, with the shifts reaching equilibrium at 300 µM, although the peaks remained broad. The remarkably large shifts of the C-terminal residues (Supplementary Fig. 13), similar to the pattern observed for $Dy^{3+}$ titration, indicate that interaction with $Ln^{3+}$ occurs mainly at the C-terminal and acidic residues. The binding affinity of this stage was calculated as $K_D = 58.6$ µM, which indicates that this is a comparatively weaker interaction than reported for various chelating molecules[36,39–41]. In the second phase, excess $La^{3+}$ led to further changes in some residues, including Leu6, Trp7, Asp14, Phe15, Leu16 and Ser18 (Fig. 3a). Typically, the Trp7 peak shifts in opposite directions before and after addition of 2,200 µM $La^{3+}$. This stage likely originates from a mineralization event, such as Lamp-1 complexation with Ln hydroxide or complex accumulation.

Previous reports indicate that many proteins involved in biomineralization contain a large number of acidic residues, which have a key role in protein function[8,12,42]. We also identified Asp to have an interesting function in our system by examining the reaction of $Dy^{3+}$ and $Nd^{3+}$ with each amino acid in Lamp-1. At pH 6.0 with low buffer concentration, the addition of Asp, but not Glu or other amino acids, decreased the pH of a solution containing $Ln^{3+}$, although insoluble particles were not generated (Supplementary Fig. 14). The acid dissociation constants ($pK_a$) of Asp and Glu side chains are 3.90 and 4.07, respectively. If $Dy^{3+}$ and $Nd^{3+}$ interactions caused the dissociation of protons from carboxyl groups, the addition of Glu would similarly decrease the solution pH. We consider that the overall structural properties of Asp, including bulkiness[43], side chain configuration and nucleophilic nature of carboxylate, may promote the generation of Ln hydroxide.

A thermodynamic analysis was conducted using isothermal titration calorimetry (ITC) to elucidate the mineralization characteristics. Titration of $Dy^{3+}$ into Lamp and LBT3 showed a large endothermic signal, which indicated that both reactions are entropy-driven (Fig. 3b, Supplementary Fig. 15 and

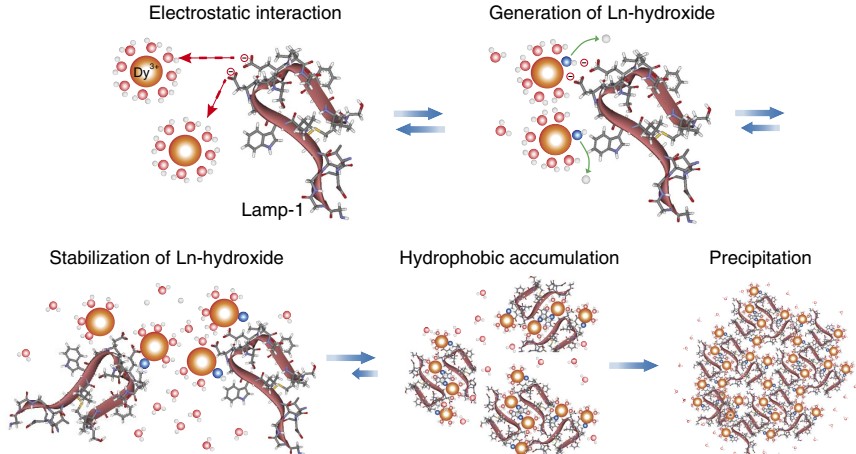

**Figure 4 | Proposed mechanism of Ln³⁺ mineralization.** Schematic illustration of the Lamp-1-induced mineralization mechanism. Lamp-1 recognizes $[Ln(H_2O)_{8-9}]^{3+}$ through electrostatic interaction. Ln hydroxide is generated by proton release near the peptide. Complexation with Lamp-1 stabilizes Ln hydroxide and prevents dehydroxylation/rehydration. Finally, the hydrophobic effect induces accumulation of Lamp-1 with Ln hydroxide complexes.

Supplementary Table 4). This result means that some constrained water molecules are freed by the following reactions. In aqueous conditions, owing to its high electric charge and large ionic radius[44,45], $Dy^{3+}$ is coordinated by 8–9 water molecules, which are released by electrostatic interaction with Lamp. This event is observed frequently for interactions of metal ions with chelator molecules[36,39]. Moreover, after the ITC measurements, complex aggregation/precipitation was observed in the Lamp reaction, which accompanies disruption of the hydration shell around the hydrophobic surface[46].

To quantify the number of protons lost upon reaction, an additional ITC experiment was conducted for the reaction of Lamp-1 and $Dy^{3+}$. The observed enthalpy change $(\Delta H_{obs})$ for the proton transfer reaction varies according to the buffer species because each buffer has a different ionization enthalpy $(\Delta H_i)$[47–49]. Consistent with this, an approximately 11-fold difference in $\Delta H_{obs}$ was observed for MES buffer and Bis-Tris buffer conditions. The slope of line calculated with equation (1) indicates a decreasing function ($-2.02$), indicating that about 2 mol equivalents of protons were transferred onto a buffer component upon complexation (Supplementary Fig. 16). From the reaction stoichiometry and the $Ln^{3+}$ equilibrium in aqueous solution, it is assumed that Lamp-1 precipitates monomeric or dimeric molecules of Ln monohydroxide.

From the above findings, the mineralization reaction cascade induced by Lamp could be explained as follows (Fig. 4). In aqueous solution, Lamp first weakly recognizes $Ln^{3+}$ through electrostatic interactions. Then, the generation of Ln hydroxide by proton release is promoted at the peptide surface owing to the acidic residues. Complexation with Lamp stabilizes Ln hydroxide, preventing dehydroxylation/rehydration. The Lamp and Ln hydroxide complex has low solubility in aqueous environments, resulting in hydrophobic accumulation. These features indicate that this phenomenon seems to mimic biomineralization in nature[12], and Lamp is distinct from previous metal-recognizing peptides[18,21,23–26], which only have binding affinity for metal crystal surfaces.

**Recovery of $Dy^{3+}$ from nature.** The unique function of Lamp could have several applications, such as selective recovery of scarce Ln in nature. We examined the fundamental ability of Lamp-1 to separate $Ln^{3+}$ from seawater, in which huge amounts of $Ln^{3+}$ species are potentially contained[50]. The addition of Lamp-1 to synthetic seawater containing $Dy^{3+}$ resulted in a cloudy mixture and particle generation was observed (Fig. 5a). ICP-OES analysis revealed that, although the concentrations of major ions in the mixture ($Na^+$, $Mg^{2+}$, $K^+$ and $Ca^{2+}$) were over 160 times higher than that of $Dy^{3+}$, the precipitation percentage of $Dy^{3+}$ was over 20-fold that of other ions (Fig. 5b). According to the amount of precipitated Lamp-1 and Dy, the reaction stoichiometry (Dy/Lamp-1) is calculated as 1.6–1.9 (Supplementary Fig. 17). Subsequently, to confirm whether our system works on molecular scaffolds, we fused Lamp-1 with two different types of macromolecules: sepharose resin and a protein. On addition of $Dy^{3+}$, the sepharose resin chemically fused with Lamp-1 showed accumulation of Dy on its surface (Fig. 5c,d and Supplementary Fig. 18). The captured Dy was easily eluted by treatment with a weakly acidic solution (pH 4.0), and the recycling ability of this resin was confirmed by repeating this capture and elution procedure five times. Similarly, genetic fusion of glutathione S-transferase (GST), which is a protein expressed in various living organisms, with Lamp-1 resulted in $Dy^{3+}$ mineralization function, besides maintaining the glutathione binding and homodimerization ability of GST (Supplementary Figs 19 and 20). In contrast, little or no Dy accumulated on the negative controls not fused with Lamp-1. These features demonstrate that Lamp-1 conjugation could graft the mineralization function to various scaffolds without impairing their intrinsic properties, which will contribute to the creation of functional materials to recover scarce Ln from seawater and industrial wastewater.

## Discussion

We identified three kinds of Hydro-$Ln_2O_3$-binding peptides from the random peptide library displayed on the T7 phage. The small structural diversity of the library used here, which covers <0.01% of the theoretical diversity, might reflect the low number of isolated peptides. However, the screening strategies (library design), including a cyclic peptide backbone, random oligo generation by the NNK method and the avidity/rebinding effect derived from the multiple display system, which are referred to in previous studies[28–34], should play an important role in successful isolation. The isolated peptides commonly contain Asp, Trp and Ser, but no basic amino acids, resulting in low pI values, although there seems to be little similarity in the motif of each Lamp. This feature should reflect the surface chemistry of Hydro-$Ln_2O_3$; the surface has a positive charge in solution at pH 7.5 (ref. 51), and Ln-OH (hydroxide) and protonated Ln-$OH_2$($+$) groups are

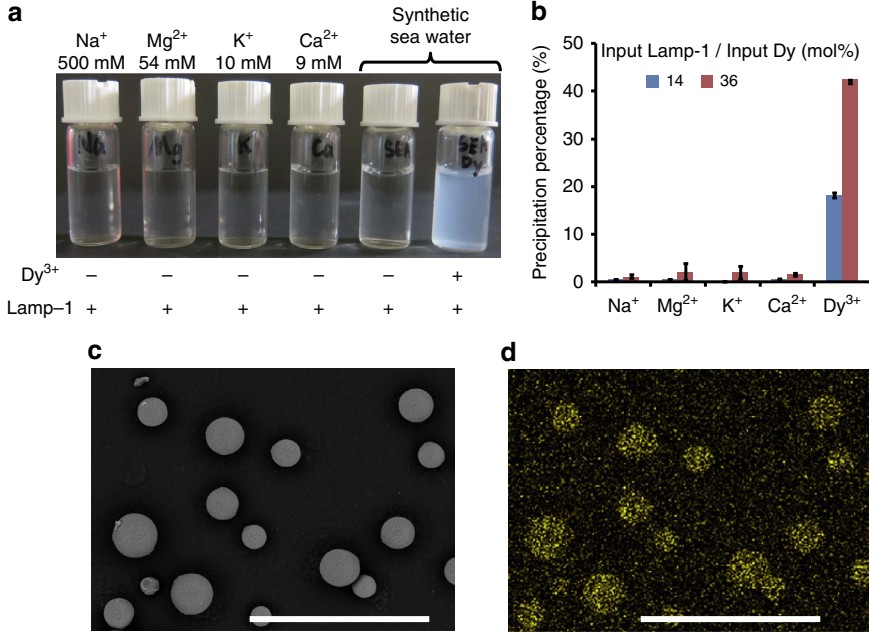

**Figure 5 | Potential of Lamp for Dy$^{3+}$ separation.** (**a**) Specific precipitation of Dy$^{3+}$ using 150 µM of Lamp-1 and 3 mM of Dy(NO$_3$)$_3$ in MES buffer (50 mM, pH 6.5). The symbols '+' and '−' in the figure indicate the presence/absence of reagents. (**b**) Lamp-1 was added at 14 and 40 mol% of Dy, and the precipitation percentage of each metal from synthetic seawater was analysed using ICP-OES. The precipitation percentage was calculated using the following equation: the amount of precipitated metal/the amount of input metal × 100. Error bars denote s.d.; $n = 3$. (**c,d**) Dy$^{3+}$ mineralization by sepharose resin conjugated with Lamp-1. SEM and EDX analyses were performed after reaction with Dy(NO$_3$)$_3$ at pH 6.1. Scale bars, 200 µm.

distributed on the particle surface, but the pattern and degree of these functional group on the particle under this condition are unregulated. Ion paring and hydrogen bonds between Lamps and the particle surface are considered to be the main absorption force, although the binding conformations of Lamps are not considered to be similar to each other. In fact, the binding strength of Lamps with Hydro-Ln$_2$O$_3$ seems to simply be related to the number of acidic residues (Supplementary Table 3).

As we expected, Lamp induces generation and precipitation of Ln hydroxide by reacting directly with Ln$^{3+}$ without any artificial reagents. Neither the La$^{3+}$-chelating peptide (LBT3)[36] nor the Ln oxide-binding peptide (RE-1)[37] causes the same phenomena (Supplementary Fig. 6). According to the stability constant of Dy monohydroxide (Log $K = 5.63$), the concentration of Dy$^{3+}$ and [Dy(OH)]$^{2+}$ are equal at pH 8.37 (Supplementary Fig. 1)[27]. At pH 6.0, Dy$^{3+}$ is at least 100 times more stable than Dy hydroxide species ([Dy(OH)]$^{2+}$, [Dy(OH)$_2$]$^+$, Dy(OH)$_3$), and, thus, this cation is the major species. Therefore, it is easily accepted that almost all Lamp first interacts with Dy$^{3+}$, although this reaction is enthalpically unfavourable ($\Delta H > 0$); the formation of an electrostatic bond is itself intrinsically enthalpically favourable, which are over compensated by the energy for removing the coordinating water molecules from Dy$^{3+}$ (refs 44,48,52). This process is also entropically unfavourable because the conformational freedom of Lamp in aqueous solution is decreased by the interaction with Dy$^{3+}$ ($-T\Delta S_{conf} > 0$, $T$: absolute temperature, $\Delta S_{conf}$: entropy change for the peptide conformation). Following this electrostatic interaction, Dy hydroxide is generated following proton release at the Lamp surface. Lamp should bind and stabilize Ln hydroxide, which prevents the reprotonation of the hydroxide anion. Moreover, binding of Lamp with Ln$^{3+}$ or Ln hydroxide achieves charge neutralization, resulting in spontaneous accumulation/precipitation. Among these reactions, the release of coordination water molecules and the disruption of the hydration structure allow these molecules to gain

translational/rotational mobility. This entropically favourable process overcompensates for the above-mentioned enthalpic disadvantages, and is considered the driving force of this Lamp-based mineralization.

The reaction cascade of Lamp-based mineralization, involving interaction with Ln$^{3+}$, stabilization of Ln hydroxide (nucleation) and complex accumulation, is quite similar to the natural phenomenon that is mediated by proteins containing many charged amino acids, especially Asp. The natural proteins, Pif 97 and Pif 80, which are involved in pearl formation, contain 14.9% Asp (Glu: 6.5%, Lys: 11.1%, Arg: 5.0%) and 28.5% Asp (Glu: 4.1%, Lys: 18.7%, Arg: 10.9%), respectively[8]. These Asp residues build acidic clusters on the protein surface and are considered to regulate crystal polymorphism[53,54]. In addition, polyaspartic acid itself has been demonstrated to affect aragonite (CaCO$_3$) and struvite (NH$_4$MgPO$_4 \cdot 6$H$_2$O) crystal formation[42,55]. The acidic clusters are assumed to function by concentrating metal ions locally to overcome the reaction barrier. As with natural proteins, Lamp contains at least one acidic amino acid, commonly Asp, which is considered to play two important roles in the mineralization event. First, electrostatic interactions bring Ln$^{3+}$ close to the peptide surface, and generation of Ln hydroxide from Ln$^{3+}$ proceeds near the Lamp surface. Subsequently, acidic residues, such as Asp, seem to promote Ln hydroxide generation. Ln$^{3+}$ has been reported to form polynuclear hydroxides, such as Ln$_2$(OH)$_2$ (refs 56,57). The high charge of Ln$^{3+}$ along with the intrinsic properties of Asp, including bulkiness[43], side chain configuration and nucleophilic nature, may promote Ln hydroxide generation, although the underlying mechanism is unknown.

The sophisticated control of metal crystal growth under reducing conditions was achieved using several artificial metal-binding peptides[18,21,25]. The key function is the ability to recognize the target surface with soft epitaxy[58,59], and artificial *in vitro* selection with highly ordered metal crystal provided the appropriate properties to these materials. On the other hand,

amorphous precipitation of metal–organic composites with no reducing reagents was demonstrated by silica- or titania-binding peptides[60–62]. The ion pairing and hydrogen bond formation, not the soft epitaxy, are important for their function; the pI values of the silica- and titania-binding peptides are 8.6–9.6 and 6.2–12.4, respectively, and they are positively charged in solution with pH > 7.5, whereas the silica and titania surfaces are negatively charged, with randomly distributed metal-OH and/or metal-O$^-$ groups[51,60,61]. Lamp, similar to the silica- and titania-binding peptides, does not control crystal growth, and the precipitates formed are amorphous in nature. The above characteristics indicate that the type of Lamp recognition is not soft epitaxy; this feature should be derived from the use of Ln$_2$O$_3$ nanoparticles with randomly distributed hydroxyl groups in peptide selection. The precise control of the metal surface chemistry and its application for peptide screening are essential for acquiring the soft epitaxial absorption ability, which might allow the identification of a peptide via ordered-composite generation.

Ln species consist of 15 elements that are difficult to separate from one another owing to their extremely similar properties. As we anticipated, the specificity of Lamp-1 for the Ln series is highly correlated with the stability constants of Ln hydroxide (Fig. 2d). There is over one order of magnitude difference in log $K$ for [La(OH)]$^{2+}$ and [Lu(OH)]$^{2+}$ (4.67 and 5.83, respectively[27]), suggesting that the limiting step of Lamp-1-based mineralization is the generation of Ln hydroxide. On the other hand, Lamp-2 and -3 showed considerable differences in the selectivity of mineralization compared with Lamp-1. According to the ICP-OES measurements, a 3–4-fold difference was observed in the reaction stoichiometry of Lamp-1 and the other Lamps (Supplementary Fig. 8). This result correlates with the difference in the $\Delta H$ values of the three kinds of Lamps because the magnitude of $\Delta H$ (dehydration) should reflected the number of Dy bound to Lamp. This difference in the recognition pattern is considered one of the reasons for the mineralization reactivity.

The remarkable advantage of Lamp-1 for direct recovery of Ln from seawater or industrial wastewater is that Ln$^{3+}$ is precipitated spontaneously without any artificial reagents (Fig. 5). This function makes the reaction almost irreversible, meaning that exchange reactions with naturally abundant metal ion species (Na$^+$, Mg$^{2+}$ and Ca$^{2+}$) do not need to be considered. It is also beneficial that Lamp does not cause specific precipitation of these metals. Furthermore, the precipitate containing Ln hydroxide could be redissolved only when the reaction conditions were changed, for example, a slight pH decrease. Lamp retains the above functions, even when conjugated on various scaffolds, which should improve the reusability of Lamp.

Finally, we demonstrated a novel mineralization concept for stabilization of an insoluble lanthanide hydroxide complex with an artificial peptide, which has potential for direct separation of Ln under mild conditions without additional energy input. Although the Lamp presented here confirmed our concept, further improvements to achieve more efficient mineralization are possible because the peptide selection was performed using a library with limited diversity. On-going optimization efforts with molecular simulations and mutated Lamps are expected to allow Ln extraction from various natural environments. This strategy could be applied for other transition metals by considering appropriate chemical reactions and bio-molecule construction, and has the potential to open a new scientific field in bio-metal chemistry.

## Methods

**Materials.** All lanthanide oxide nanoparticles and lanthanide nitrates were purchased from Sigma-Aldrich (St Louis, MO). Lanthanide oxide nanoparticles (Ln$_2$O$_3$-NP) were washed with a 1:1 mixture of methanol and acetone, and then ultrasonicated for 10 min using a Bioruptor UCD-250 ultrasonicator (Cosmo Bio, Tokyo, Japan). After washing twice with isopropanol, Ln$_2$O$_3$-NP were dispersed in Tris-buffered saline (TBS; 50 mM Tris, 500 mM NaCl, pH 7.5) containing 0.1% Tween 20, and then washed twice more with TBS buffer. After each washing step, the sample was centrifuged at 6,000 r.p.m. for 10 min. Lanthanide trihydroxides were purchased from Mitsuwa Chemicals (Osaka, Japan) and Nippon-Yttrium (Fukuoka, Japan). Simplified synthetic seawater was prepared according to a previous report[63]. In brief, 453 mM sodium chloride, 34 mM magnesium chloride, 17 mM magnesium sulfate, 9.6 mM potassium chloride and 10 mM calcium sulfate were dissolved in distilled water.

**FT-IR spectroscopy.** The FT-IR spectra were recorded on a Nicolet Avatar 360 FT-IR spectrometer (Thermo Scientific, Waltham, MA). To remove the surface adsorbed water, dysprosium oxide nanoparticles (Dy$_2$O$_3$-NP) and neodymium oxide nanoparticles (Nd$_2$O$_3$-NP) were incubated at 80 °C for 2 h under vacuum conditions. The spectra were recorded by accumulating 100 scans from 600 to 4,000 cm$^{-1}$. The Fourier transformation of all spectra was performed using Omnic E.S.P. 5.1 software.

**Phage display and peptide screening.** The T7 phage libraries displaying SCX$_{9-12}$CS random peptides, where X represents the randomized amino acids generated by mixed oligonucleotides on a DNA template[32], were constructed using the T7Select 10-3b system (Merck Millipore, Billerica, MA). The T7 phage displays an average of 5–15 copies of the peptide on the surface of the phage particle. Treated Ln$_2$O$_3$-NP (500 µg) were added to the constructed SCX$_{9-12}$CS libraries (5 × 10$^{10}$ plaque forming units; pfu), and then incubated for 1 h at room temperature. Subsequently, Ln$_2$O$_3$-NP were washed 5–20 times with TBS buffer containing 0.1–0.3% Tween 20. For proliferation of T7 phage bound to the surface of Ln$_2$O$_3$-NP, 10 ml of *Escherichia coli* BLT5403 (Merck Millipore) proliferated to the log phase was mixed with the nanoparticles and incubated at 37 °C by shaking until bacteriolysis. After bacteriolysis, the phages were recovered from the culture supernatant according to the manufacturer's instructions, and the recovered phage solution was used for the next round of screening.

**Identification of peptide sequences.** DNA fragments inserted in the vector of the monoclonal T7 phage were amplified by PCR using PrimeSTAR Max DNA polymerase (Takarabio, Shiga, Japan). The PCR reaction was initiated at 98 °C for 3 min, followed by 30 cycles at 98 °C for 10 s, 55 °C for 10 s and 72 °C for 5 s using a Veriti 96-well thermal cycler (Applied Biosystems, Waltham, MA). The oligonucleotide primers used in this reaction had the following synthetic sequences (Eurofins Genomics, Tokyo, Japan).

T7 forward sequencing primer: 5′-GGA GCT GTC GTA TTC CAG TC-3′ (20 mer)

T7 reverse sequencing primer: 5′-AAC CCC TCA AGA CCC GTT TA-3′ (20 mer)

The peptide sequence was determined through analysis of the DNA sequence using a Genetic Analyzer 3130 and BigDye Terminator v3.1 cycle sequencing kit (Applied Biosystems).

**Detection of phage or peptide binding to Dy$_2$O$_3$-NP.** Selected phage mixtures were mixed with 500 µg of Ln$_2$O$_3$-NP and incubated for 1 h at room temperature. After washing five times with TBS-T buffer (50 mM Tris, 150 mM NaCl, 0.1% Tween 20, pH 7.5), the number of remaining phages on Ln$_2$O$_3$-NP was determined according to the manufacturer's instructions.

The synthetic peptides were mixed with 500 µg of Ln$_2$O$_3$-NP using the same condition as in the above protocol. After washing five times with TBS-T buffer, horseradish peroxidise-conjugated streptavidin (Novagen) diluted (1:5,000) in TBS-T buffer containing 0.5% BSA (bovine serum albumin) was added to the sample and incubated for 1 h. Each sample was washed five times with TBS-T, followed by addition of the substrate 3,3′,5,5′-tetramethylbenzidine (Wako Pure Chemical Industries, Osaka, Japan). In each washing step, centrifugation was performed at 6,000 r.p.m. for 10 min. After stopping the reaction with 1 N HCl, the absorbance of each sample was measured at 450 nm using a microplate reader (Molecular Devices Spectra Max Plus 384, Sunnyvale, CA).

**Peptide synthesis.** All peptides were synthesized by a custom peptide synthesis service (Scrum, Tokyo, Japan). The synthetic peptides were prepared by solid phase synthesis using the 9-fluorenylmethyloxycarbonyl (Fmoc) group. Some peptides were *N*-terminally biotinylated using a Gly-Gly-Gly spacer from the g10 protein of T7 phage. After removal of the protecting groups from the 4-hydroxymethyl phenoxymethyl polystyrene (HMP) resin, the peptides were mildly oxidized to form intramolecular disulfide bonds. The generated disulfide-constrained peptides were purified by reverse phase high performance liquid chromatography (HPLC). After lyophilization, the peptides were dissolved in the appropriate buffers and, after centrifugation, used for assays. The purity of these peptides and the formation of disulfide bonds were confirmed by HPLC–mass spectrometry.

**Mineralization of lanthanide ions.** Peptide solutions (20–300 μM) were mixed with lanthanide nitrate (Ln(NO₃)₃) solutions (up to 3 mM) and incubated at room temperature. All experiments were carried out at room temperature for 0–25 h in buffer (50 mM MES, pH 6.0 or 50 mM HEPES, pH 6.8) containing 0–5% (v/v) dimethyl sulfoxide (DMSO).

**Reverse phase chromatography.** Reverse phase chromatography was performed using an Inertsil ODS-3 column (GL Sciences, Tokyo, Japan) equilibrated with 5.0% acetonitrile at a flow rate of 1 ml min⁻¹. Precipitates generated during the mineralization experiment were dissolved in equilibration buffer and injected into the column. Peptide elution was controlled by the gradient of acetonitrile (up to 50%) and monitored by absorbance at 215 nm. All solutions contained 0.1% trifluoroacetic acid.

**SEM–EDX analysis.** SEM–EDX analysis was performed using a TM3000 microscope (Hitachi High-Technologies, Tokyo, Japan) operated at 15 keV. The particles generated during the mineralization experiments were separated by centrifugation at 15,000 r.p.m. for 10 min. The supernatant was removed and the precipitate was washed with pure water. The samples dispersed in water were dropped onto carbon tape and dried under atmospheric conditions before obtaining the images.

**TEM–EDX analysis.** TEM–EDX analysis was conducted using a JEM-2100F microscope (JEOL, Tokyo, Japan) operated at 200 kV. The particles generated during the mineralization experiments were separated by centrifugation at 15,000 r.p.m. for 10 min. The supernatant was removed and the precipitate was washed with MES buffer (10 mM, pH 6.0). The samples dispersed in buffer solution were dropped onto carbon-coated copper grids and dried under atmospheric conditions before obtaining the images.

**ICP-OES analysis.** The concentration of metal ions was analysed using inductively coupled plasma-optical emission spectrometry (ICP-OES, CIROS 120, Rigaku, Tokyo, Japan). Synthetic lanthanide ion mineralization peptides (Lamps) (100–300 μM) with no biotin were added to metal ion solutions and incubated for 20 h with continuous stirring at room temperature. After precipitation of the generated particles by centrifugation at 6,000 r.p.m. for 10 min, the supernatant was recovered to measure the ion concentration. All sample solutions were diluted with ∼1.3% (w/w) HCl (Wako Pure Chemical Industries) and adjusted to 50 g with pure water. Calibration was carried out using standard solutions of each metal (Kanto Chemical, Tokyo, Japan) before the measurements. The 1.3% (w/w) HCl solution was used as a blank and to rinse the instrument after measuring the most concentrated standard solution. The reported values are the averages obtained from three individual measurements.

**XAFS analysis.** The XAFS measurements were performed at beamline BL33XU of SPring-8 (Hyogo, Japan). A Si (111) channel-cut crystal cooled with liquid nitrogen was used to monochromatize the X-rays. Higher harmonics of the X-rays were rejected using a pair of Rh-coated Si mirrors. The beam size at the sample position was 0.5 mm (H) × 0.5 mm (W). The X-ray intensities were monitored using ionization chambers filled with a N₂–He mixture (30%/70%) for the incident beam and an Ar–N₂ mixture (15%/85%) for the transmitted beam. Dy $L_3$-edge and Nd $L_3$-edge XAFS spectra were collected in transmission mode over 15 min. All XAFS data analysis was performed using the Athena software[64].

**NMR spectroscopy.** NMR experiments were carried out at 298 K, using a Bruker 900 MHz Avance III HD spectrometer equipped with a TCI cryoprobe. Standard 5 mm NMR tubes were used for the measurements. The samples were prepared in 10 mM MES-$d_{13}$ buffer (H₂O:D₂O:DMSO-$d_6$ = 85:10:5, pH 6.0, Sigma-Aldrich), and the concentration of synthetic peptides was adjusted to 50 μM. The lanthanides were added as the nitric acid salts (5 mM stock solutions in the same buffer), and the samples were allowed to equilibrate for at least 10 min before measurement. The ¹H NMR spectra of the all peptides were assigned using the standard homonuclear two-dimensional NMR methodology[65]. Total correlation spectroscopy was conducted with a mixing time of 60 ms, and nuclear Overhauser effect spectroscopy with a mixing time of 300 ms. The titration experiments were performed by adding Ln ion solutions to the peptide solutions. 1D and total correlation spectroscopy spectra were measured at each titration step to trace the chemical shift changes. All NMR data were processed and analysed with Topspin 3.1, NMRPipe[66] and Sparky[67].

**ITC analysis.** The thermodynamic parameters of binding between Ln³⁺ and the peptides were analysed at 25 °C by ITC (MicroCal VP-ITC, MicroCal PEAQ-ITC, Malvern, Worcestershire, UK). The synthetic peptides, dysprosium nitrate hexahydrate (Dy(NO₃)₃), and neodymium nitrate hydrate (Nd(NO₃)₃) were each dissolved in 50 mM MES buffer or Bis-Tris buffer (pH 6.0). The peptide solution was placed in the calorimeter cell, and the Dy(NO₃)₃ or Nd(NO₃)₃ solution was loaded into the syringe injector. The titrations were performed to a 100:1 ratio with final concentrations of Ln(NO₃)₃ between 5 and 10 mM and 50–100 μM of Lamp

and RE-1. In the case of LBT3, the titrations were performed to a 10:1 ratio with 500 μM of Dy(NO₃)₃ and 50 μM of peptide. The experiments included 29 injections, whereby an initial 2 μl injection was used to account for dilution of the syringe, and the remaining injections were 10 μl with a 200 s delay between each injection. The effect of Ln³⁺ dilution in the cell was calculated by subtraction of titration data for a blank, which consisted of titrating Ln³⁺ into a buffer solution. Each binding parameter, including stoichiometry (N), association constant (K) and binding enthalpy (ΔH), was calculated using the ITC Origin Analysis Software ver. 7.0 (Malvern). For the samples that did not show a sigmoidal response owing to a low K value, the thermodynamic parameters were calculated by fixing the stoichiometry to 1.0 according to the manufacturer's instructions. The number of protons released was calculated using the following equation[47].

$$\Delta H_{obs} = \Delta H_{bind} + (\Delta n)\Delta H_i, \qquad (1)$$

where $\Delta H_{obs}$ is the enthalpy change observed in the ITC experiment, $\Delta H_{bind}$ is the enthalpy change without a buffer effect, $\Delta n$ is the number of protons transferred and $\Delta H_i$ is the ionization enthalpy of each buffer.

**Dy³⁺ mineralization with Lamp-1-immobilized sepharose resin.** Lamp-1 was immobilized on NHS-activated Sepharose 4 Fast Flow medium (GE Healthcare) according to the manufacturer's instructions. In brief, 100 μl of sepharose resin was harvested in a tube, and the supernatant was removed carefully. Immediately after mixing 1 ml of cold HCl (1 mM) with the resin, Lamp-1 dissolved in 0.2 M NaHCO₃ (Wako Pure Chemical Industries) containing 0.5 M NaCl (pH 8.2) was added. The reaction was incubated at 4 °C overnight. The amount of immobilized peptide was estimated by analysing the decrease of peptide concentration in the supernatant. Tris (hydroxymethyl) aminomethane (Tris) was immobilized as a control. The prepared sepharose resins were mixed with 10 mM Dy(NO₃)₃ in 50 mM MES buffer (pH 6.1) and incubated for 24 h at room temperature with continuous stirring. After incubation, each resin was washed with MES buffer two times and analysed by SEM or EDX after drying.

**Construction of the GST-Lamp-1 expression vector.** T7 phage DNA carrying the *Lamp-1* gene was used as a template, and the following *Lamp-1u* and *Lamp-1d* primer set was applied for amplification of the *Lamp-1* gene. The PCR products were digested by the *Spe* I and *Hin*d III restriction enzymes, and then ligated into the pET42a Expression Vector (Novagen), which was digested by the same restriction enzymes. The constructed vectors were transferred to the ECOS competent *E. coli* BL21 (DE3) strain (Nippon Gene, Tokyo, Japan) according to the standard protocol. For the control experiments, GST with no Lamp-1 was constructed using the following *GST-STOP* and *Lamp-1d* primer set. The primer gene *GST-STOP* contains a stop codon just after the restriction enzyme *Spe* I.

*Lamp-1u*: 5′-ATAATATGAACTAGTTCAGGTGGAGGTTCGTGTTTGT-3′
*Lamp-1d*: 5′-ACTATCGTCGGCCGCAAGCTTTTAGCT-3′
*GST-STOP*: 5′-ATAATATGAACTAGTTCATAAGGTGGAGGTTCGTGTTTGT-3′

**Expression and purification of the recombinant protein.** Transgenic *E. coli* was cultivated in 100 ml of LB medium containing 50 μg ml⁻¹ ampicillin with shaking at 160 r.p.m. at 37 °C for 20 h. After cultivation, the cells were collected by centrifugation and were washed with 20 mM phosphate buffer (pH 7.4) containing 0.5 M NaCl. A crude protein mixture was prepared using an xTractor buffer kit (Clontech Laboratories). Protein purification was performed using GSTrap HP columns (GE Healthcare), and the protein was then concentrated using Amicon Ultra-15 centrifugal filter units (10k MW, Merck Millipore). The prepared samples were desalinized with HiTrap desalting columns (GE Healthcare). The concentrations of the purified proteins were calculated from their respective absorbances at 280 nm and molar extinction coefficients (Bio Spectrophotometer, Eppendorf, Hamburg, Germany).

**SDS–PAGE and Western blotting.** Sodium dodecyl sulfate-polyacrylamide gel electrophoresis (SDS–PAGE) was performed on 4–20% gradient polyacrylamide gels (TEFCO, Tokyo, Japan) according to the general protocol. Proteins were stained with the GelCode Blue Stain reagent (Thermo Scientific, Waltham, MA). The molecular weights of the proteins were estimated using standard protein markers (Precision Plus Protein unstained protein standards) containing *Strep*-tagged recombinants (Bio-Rad, Hercules, CA). After electrophoresis, the proteins were transferred onto a polyvinylidene difluoride membrane using an iBlot system (Life Technologies, Carlsbad, CA) and blocked with 5% skim milk (Wako Pure Chemical Industries) in TBS buffer (50 mM Tris-HCl, pH 7.4, 150 mM NaCl) for 2 h. After washing with TBS-T, the membranes were incubated with Anti-GST-tag HRP DirectT (Medical & Biological Labs, Nagoya, Japan) for 1 h. SuperSignal West Femto Maximum Sensitivity Substrate (Thermo Scientific) was used as a detection reagent.

**Data availability.** Data from the experiments presented in this study are available from the corresponding author upon request.

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

## Acknowledgements

We gratefully acknowledge S. Higuchi, M. Okumura, Y. Fujiyoshi, M. Ohashi and Y. Hosokawa at Toyota CRDL for technical assistance with all data collection. We also thank S. Kosaka and Y. Akimoto at Toyota CRDL for ICP-OES and TEM–EDX analysis. The XAFS measurements were performed at the BL33XU of SPring-8 with the approval of the Japan Synchrotron Radiation Research Institute (JASRI) (Proposal No. 2014B7026). The NMR experiments were performed with the approval of the RIKEN NMR Facility (Proposal No. 14-500-047). The XAFS and the NMR experiments were supported by grants and subsidies for 'Open Advanced Research Facilities' from the Ministry of Education, Culture, Sports, Science and Technology, Japan.

## Author contributions

T.H., T.T. and N.I. designed the experiments and wrote the manuscript. T.N. and H.T. performed the XAFS experiments. A.M. and F.H. performed the NMR experiments. T.H. prepared the samples for the XAFS and NMR experiments, and performed all of the other experiments. All authors discussed the results and commented on the manuscript.

## Additional information

**Competing interests:** Patent applications have been filed for the technology described in this publication.

