## [Peer Review File · Nature Communications]

Reviewers' comments:

Reviewer #1 (Remarks to the Author):

In this study Hatanaka and coworkers use phage display to isolate three "Lamp" peptides binding to lanthanide (Ln) trioxides, demonstrate that the synthetic peptides can induce precipitation of Ln³⁺ ions, and use of a variety of techniques (including XAFS, NMR, ITC) to propose a mechanism for this activity - namely that acidic residues, and especially Asp, attack the hydration shell to generate and stabilize Ln-hydroxides that eventually aggregate. The authors also show that the Lamp peptides are suitable for precipitating Ln from synthetic seawater; that some selectivity exists in Ln precipitation depending on the peptide used; and that the Lamp1 peptide is functional for Ln precipitation when conjugated to sepharose or fused to glutathione-S-transferase.

This comprehensive study (there is a large amount of supplementary data) should be of interest to the readership of Nature Communications. However, I have a number of questions, comments and suggestions for the authors.

- The title claims "rationally designed mineralization" but I am at a loss at finding what is rationally designed.
- The authors make use of disulfide-constrained "Lamp" peptides in all experiments. Are similar results obtained with linear peptides?
- What does "which is based on known metal-binding sequence" mean (line 76)? The library seems random to me.
 - Why were these 3 Lamp peptides picked? How many peptides were obtained through the screen and how different were they from one another? The authors mentioned that they screened on both Dy₂O₃ vs Nd₂O₃. Were different peptides obtained on the two materials?
 - There is no consistency in buffers (MES, HEPES) and pH (6.0, 6.1, 6.5, 6.8) for the precipitation experiments of Figs. 2, S4, S5, 5
- A scrambled Lamp-1 peptide or unrelated peptide of identical size should be included as a control in the experiment of Fig. 2a.
 - In the same Fig, and more importantly, how long and at what temperature are these precipitation reactions performed. Is there an influence of time on the amount of materials precipitated as suggested by the experiments of Fig. S6 (performed in a different buffer and at a different pH).
 - Why is the morphology of the precipitate visualized by SEM so different when Lamp1 is used (Fig. 2b) compared to Lamp 2 or Lamp 3 (Fig S7c/d). Also why is the underlying Cu grid signal absent in the EDX spectra of Fig. S7c/d?
- How can error bars be provided for n=2 in Fig. 2d and S9?
- Could the authors speculate on why there is such an abrupt transition in the ability of Lamp1 to precipitate Tb but not lower atomic mass Ln in Fig. 2d (and also S9)?
- I am confused about the acidification experiments. Aren't these solutions buffered at 50 or 100 mM?
- Why do the La³⁺ NMR experiments require titration to 2 mM while 2 μM suffice for Dy³⁺?
- Why is Lamp3 so much more promiscuous than Lamp1 and Lamp2 in terms of Ln precipitation? How is this consistent with the proposed mechanism considering that Lamp1 and Lamp3 have the same numbers of D (Lamp3 has in fact one more acidic residue than Lamp 1) and Lamp 2 has a single D and no E? Does poly-D causes Ln precipitation?
- In the same vein, the authors invoke an entropic argument to contrast the behavior of Lamp1 and LBT3 but the difference between the two energies is less than 2 kcal/mol. In addition, Lamp2, which seems to function much like Lamp1 in Ln precipitation has a 2-fold lower entropic component. Some further discussion of these topics would be useful to the reader.
- It would be interesting for the authors to comment on the economics of the precipitation process in seawater (1 peptide is consumed to precipitate at best two Ln³⁺) and on whether or not the Ln

concentrations used in the experiments of Fig. 5 (3 mM Dy³⁺) are realistic. Also, can lanthanides captured on the Lamp1-derivatized sepharose be recovered and the resin regenerated to a functional state?

- For how long and at what T was the precipitation experiment of Fig. S22 conducted? Is there a difference at longer time points?

Minor comments

- The term aptamers is usually used for nucleic acids not peptides. Typically such peptides are referred to as solid, materials or inorganic binding peptides.

- The authors should qualify the title of Fig. 4 by calling it a "proposed mechanism"

Reviewer #2 (Remarks to the Author):

In this creative and interesting work, the authors have developed a protocol that may have practical applications for the selective removal of lanthanides from aqueous solution. Using phage display and lanthanide-oxide nanoparticles with surface hydroxides, they have selected for peptides that bind lanthanide ions as their hydroxide species. Whether by design, or luck, these peptides form lanthanide complexes with low solubility that can be isolated from aqueous solution. Extending this method to selectively remove Ln³⁺ ions, they show that these peptides are still effective when tethered to a separable resin or protein. Thus, this manuscript would seem to describe the novel results expected for publication in a Nature journal.

My enthusiasm for this manuscript, however, is tempered by some deficiencies, that may or may not be readily addressed.

First, the extensive results in the Supplementary Information (22 figures and 5 tables) suggest this is no early breakthrough, but a very well developed project (over a dozen physical methods are used to characterize the lanthanide-peptide complexes), and Nature Communications may not be the most appropriate venue for this work.

Second, in spite of some mechanistic insight (e.g., NMR characterization of the peptide complexes with paramagnetic Dy³⁺ and diamagnetic Ln³⁺), some of the results seem to be rather empirical and some of the models suggest that fundamental coordination chemistry has not been considered for the metal-protein interactions (e.g., the chelate effect is likely to be important here with multiple peptide carboxylates). A number of questions remain about the mechanism for the selective precipitation of lanthanide ions, such as the origin of the lanthanide selectivity of the peptides in Figure S10 and the discrepancy between the data in Figure S9 and S10c.

Third, I am concerned about the naive use of isothermal titration calorimetry (ITC) and over interpretation of the ITC data. A fixed stoichiometry of $n = 1$ was used to fit the data for Dy³⁺ binding to Lamp-1 and -2 and Nd³⁺ binding to Lamp-3, when the data in Figure S18 for Dy³⁺ binding to Lamp-1 show the equilibrium ratio to be closer to $n = 2$ (1.62-1.87). More worrisome, Figure 4 suggests four steps to the formation of lanthanide-peptide precipitate (binding, de-protonation, aggregation, precipitation), and there is no sense of how many of these occur during the ITC measurement. If all of them occur with each injected aliquot, then the experimental thermodynamic data are the sum for the overall process and the interpretation of individual contributions is suspect. For example, what does an experimental K_D really represent if it included the irreversible precipitation of an insoluble species? Comparison of results with Lamp-1 and LBT3 may be instructive, but the difference between -12.7 and -10.9 kcal/mol for the value of $-T\Delta S$ has little molecular significance. It is possible to use ITC to quantify the number of protons lost/gained upon complex formation, and I encourage the authors to exploit this capability for molecule insight on their system.

Finally, certain word choice suggests a weak understanding of the molecular processes involved in this phenomenon. For example, the word "rate", which includes a time component (kinetics), is used in

several places to describe an equilibrium amount, where the appropriate term would be "extent" or "percentage". The first paragraph of the Discussion has a description that mixes the concepts of kinetics ("slower", "immediately", "rapidly") and thermodynamics ("stability constant", "binding strength"). The authors promote a new term, "interruption of chemical equilibrium", to describe a simple phenomenon whereby La^{3+} coordination by the (chelating) peptide shifts a solution equilibrium to stabilize an insoluble lanthanide-hydroxide complex. I feel that the author's term has a misleading connotation.

Reviewer #3 (Remarks to the Author):

(1) The authors introduce lanthanide ion mineralization aptamers (Lamp) for the direct extraction of rare earth ions from solutions by mineralization. The peptides are derived from phage display and disrupt the chemical equilibrium between soluble Ln^{3+} ions and the Ln hydroxide species to afford precipitation, similar to techniques applied for other soluble metal ions and oxides/hydroxides. The literature is appropriately referenced; some reference to simulation studies that characterized the binding mechanisms more specifically are missing and the understanding/comparisons of mechanisms can be refined (e.g. JACS 2012, 134, 6244; Chem. Soc. Rev. 2016, 45, 412).

(2) The role of Cys in the chosen peptides is not clear - Cys is somewhat attracted to metal, although mostly to elemental metals that are not present here (see e.g. Soft Matter 2011, 7, 2113; JACS 2013, 135, 11048). The biopanning approach itself appears to be carried out with care and lead to strongly binding sequences. It may be hard to say that these would be the strongest possible binding sequences, however, as the phase library for a 12 peptide covers only 10^9 out of $20^{12} \sim 10^{16}$ peptides (this is less than one-millionth of possibilities). In fact, the reported peptides are even up to 16 amino acids long (Lamp no 3, p. 4). A comment on the limitations would be suitable here and not at all affect the credibility or impact of the manuscript.

(3) Typographical: line 122: "chemical shits" into "chemical shifts"

(4) The interpretations of the binding mechanism on the basis of NMR chemical shifts are carefully performed and helpful to ascertain dominant interactions. Isothermal titration adds valuable information as to the pH dependence of the mineralization point and the general prevalence of protonated/deprotonated species.

(5) On p. 8 it may be noted that the Lamp peptides are similar to oxide recognizing/forming peptides such as silica or titania binders. They have virtually nothing in common with metal binding peptides as there is no elemental metal in the process here. The authors might want to clarify that metal-binding peptides recognize metal surfaces by soft epitaxy, not by ion pairing or electrostatic interactions (see Chem. Soc. Rev. 2016, 45, 412; or original studies such as the ref. in Soft Matter 2011 above, Nano Lett. 2013, 13, 840; Adv. Funct. Mater. 2015, 25, 1374).

(6) p. 7-9: The net entropy increase upon peptide binding to oxide/hydroxide nuclei due to freed water molecules is consistent with earlier studies of peptide binding to silica and metal surfaces (JACS 2009, 131, 9704; Chem. Mater. 2014, 26, 5725).

(7) p. 9-11: Again, I would like to remind the authors not to confuse metal nanocrystal growth with oxide growth. The acting peptide recognition, surface chemistry, and growth mechanisms are fundamentally different (soft epitaxy independent of pH versus ionic and pH sensitive acid-base chemistry). To explain the extraction of metal ions with the chosen aptamers, the recognition of aptamers on elemental metal substrates is only marginally relevant (see comment 5 above).

(8) Comments on methods and characterization: This manuscript is a very accurately informed account of the stability of metal ions, hydroxides, and mineralization of lanthanide metal ions with peptides. Surface characterization of Dy_2O_3 and Nd_2O_3 nanoparticles by FTIR shows the presence of OH surface termination in solution. Biopanning and turbidity measurements are well documented. TEM, SEM, and XAFS data characterize the elemental composition of precipitates; the pH sensitivity was tested; NMR chemical shifts were measured and meticulously assigned, including 2D TOCSY

spectra and clear identification of peptide residues in contact with metal ions (such as for Dy³⁺). Binding constants are calculated including clear designation of error bars. Thermodynamic analyses of the reactions are included. EDX was employed to verify the accumulation of metal ions in the precipitates. Mineralizing protein aptamers and originating phage DNA were analyzed. Binding strengths and error bars of peptides to Ln³⁺ and hydroxylated Ln₂O₃ are reported in exceptional precision - the study appears by far more extensive and meticulously performed than many others reported previously for other substrates.

(9) The TOC graphic is appropriate. The movie is somewhat plain and simple, but conveys the message.

Summary: Scientifically, this paper would in my opinion rank in the top 5%. The relevance of the lanthanides may be somewhat debated, although I concur with the authors that the specialty elements already find many niche applications and will remain in demand for the foreseeable future. Given the high cost, recovery by mineralization, or at least knowing possible binding constants and pathways to mineralize dissolved ions are important knowledge, and this manuscript clearly advances the field by providing example protocols and valuable reference data. I recommend the authors to perform suggested (minor) revisions and address all mentioned concerns.

Response to Reviewer #1

We thank Reviewer #1 for pointing out important issues in our manuscript. In this revision, we have made careful corrections according to the reviewer's suggestions. Moreover, we have revised the results by performing additional examinations. Our detailed responses are as follows.

- 1) The title claims "rationally designed mineralization" but I am at a loss at finding what is rationally designed.

We appreciate the reviewer's comment about the title. There are two reasons why we use the term "rationally designed mineralization" in the title. First, the concept of this study is to rationally target metal ion mineralization. The most important point of this study is that we paid attention to the stability constants of metal hydroxides to generate selective precipitation from a solution of mixed metal ions. We believe that this concept of rational mineralization can also be applied to other metal ions. Second, the peptide library construction was rationally designed. To isolate the target peptide efficiently, we constructed an elaborately designed cyclic peptide library based on other inorganic binding peptides. In the original manuscript, our explanation about the construction of the peptide library was insufficient. We have added a detailed description regarding the construction of the peptide library (see line 74–80).

- 2) The authors make use of disulfide-constrained "Lamp" peptides in all experiments. Are similar results obtained with linear peptides?

We have performed experiments using linear forms of Lamp-1, which was designed by substituting two Cys for Ala. Reaction with this linear peptide also causes precipitation; however, it showed decreased reactivity compared with the cyclic peptide. In the construction of the peptide library, we adopted the findings of the previous studies concerning conformational entropy (see line 76–77 and References 28–30). Therefore, it is possible that increasing the conformational entropy would affect Dy^{3+} mineralization.

- 3) What does "which is based on known metal-binding sequence" mean (line 76)? The library seems random to me.

As mentioned in response 1), our explanation about the construction of the peptide library was insufficient. Previous reports indicate that inorganic binding peptides tend to contain more hydrophilic amino acids than hydrophobic amino acids (References 18, 21, and 22). With reference to these studies, we constructed a random peptide library with a tendency to include 56.25% hydrophilic amino acids using the NNK codon. Therefore, even if the diversity of our peptide library was not necessarily high, peptides with the expected function could be selected efficiently. We added a detailed explanation regarding the tendency to include hydrophilic amino acids in the construction of the peptide library (see line 77–80 and Reference 32).

- 4) Why were these 3 Lamp peptides picked? How many peptides were obtained through the screen and how different were they from one another? The authors mentioned that they screened on both Dy₂O₃ vs Nd₂O₃. Were different peptides obtained on the two materials?

Lamp-1 and -2 were isolated from the screening against Dy₂O₃ and Lamp-3 was isolated against Nd₂O₃. By sequencing analysis after the five rounds of bio-panning, several redundant peptide sequences were confirmed in the clones picked up randomly (Lamp-1 and -2: 2/24 and 3/24 clones, respectively, Lamp-3: 11/88 clones). Additionally, the three kinds of phage displaying Lamp showed higher binding ability for each nanoparticle than the other clones. Therefore, we selected these three peptide sequences as the candidates.

We picked up a total of 112 phage clones and obtained 99 peptide sequences (3 kinds of Lamp and 96 other clones). The three Lamps have similarities in amino acid content, containing at least one Asp and no basic residues, although it was hard to find any similarities between Lamp and the other 96 peptides. We have added an explanation regarding the above the bio-panning results (see line 83–94).

- 5) There is no consistency in buffers (MES, HEPES) and pH (6.0, 6.1, 6.5, 6.8) for the precipitation experiments of Figs. 2, S4, S5, 5

There are two reasons why we examined these systems in different buffer and pH conditions.

1. To demonstrate that the mineralization reaction does not occurred only under very

specific conditions.

2. The buffering effect of HEPES is in the range of pH 6.8 to 8.2. However, the lanthanide species are easily hydroxylated above pH 7.5, which causes precipitation. To demonstrate Lamp function but avoid this pH effect, we adjusted the pH to 6.8 when using HEPES buffer.

- 6) A scrambled Lamp-1 peptide or unrelated peptide of identical size should be included as a control in the experiment of Fig. 2a.

According to the reviewer's suggestion, we have performed an additional experiment using newly synthesized peptide (NC-1), which has the same size as Lamp-1 but an almost completely unrelated sequence. We have added these results as the new data in this revision (see line 99-100, Fig. 2a, Supplementary Fig. 3, and Supplementary Table 2).

- 7) In the same Fig, and more importantly, how long and at what temperature are these precipitation reactions performed. Is there an influence of time on the amount of materials precipitated as suggested by the experiments of Fig. S6 (performed in a different buffer and at a different pH).

We performed all precipitation experiments at room temperature at various reaction times. The detailed experiment conditions were revised in the text (see line 361, and line 78-83 in the Supplementary Information). As shown in Supplementary Movie 1 and Supplementary Figure 4, the generation of this precipitate occurred almost instantaneously at the time of mixing Lamp with Ln^{3+} , and the picture in Fig. 2a was obtained just after the reaction (<3 min). As the reviewer pointed out, the amount of materials precipitated increases as time proceeds. In this revision, we have added new figures in Supplementary Figure 5 (c, d) that show the system at long times. Additionally, we added a fitted curve that plots the speed of the increasing turbidity as a function of Dy^{3+} concentration to Supplementary Fig. 4(b).

- 8) Why is the morphology of the precipitate visualized by SEM so different when Lamp1 is used (Fig. 2b) compared to Lamp 2 or Lamp 3 (Fig S7c/d). Also why is the underlying Cu grid signal absent in the EDX spectra of Fig. S7c/d?

We do not understand yet what causes the difference in the morphology of the

precipitate. We hypothesize that differences in solubility, structure, and isoelectric point of the peptide influence the morphology of the precipitate.

In the SEM experiment, because we used carbon tape for mounting the samples (see line 375–377), Cu signals were absent from the EDX analysis in Supplementary Fig. 7.

- 9) How can error bars be provided for $n=2$ in Fig. 2d and S9?

We apologize for this oversight. These data were obtained from three independent examinations, not two. We revised the legends of Fig. 2d and Supplementary Fig. 8 according to the reviewer's instructions.

- 10) Could the authors speculate on why there is such an abrupt transition in the ability of Lamp1 to precipitate Tb but not lower atomic mass Ln in Fig. 2d (and also S9)?

As shown in Supplementary Figure S1(b), there is a conspicuous gap in Log K between Tb^{3+} and Gd^{3+} . The mineralization ability of Lamp-1 was highly dependent on the stability constant of Ln hydroxide (Log K). Therefore, this result seems to reflect the difference in Log K .

- 11) I am confused about the acidification experiments. Aren't these solutions buffered at 50 or 100 mM?

In the acidification experiments shown in Fig. 3(b) in the original manuscript, we used 0.1 mM MES buffer because it is too difficult to adjust the initial pH without buffer. In this revision, we have moved the data to the Supplementary Fig. 14a, and we have added the detailed conditions for this analysis in the Figure Legend (see line 175–178 in the Supplementary Information).

- 12) Why do the La^{3+} NMR experiments require titration to 2 mM while 2 μM suffice for Dy^{3+} ?

Paramagnetic metal ions broaden the NMR spectra strikingly, even at low concentrations. In fact, titration of Dy^{3+} , which is a paramagnetic element, broadens the chemical shift of Lamp-1, even at a low concentration (2 μM), and further titration to $\sim 50 \mu M$ causes almost all of the peaks to disappear. The use of a low

concentration of Dy is therefore appropriate to demonstrate which residues in Lamp-1 first contact Dy³⁺. On the other hand, La³⁺ is diamagnetic and its titration up to 2 mM allows observation of the shift in the NMR spectrum, which is derived from the interaction of La³⁺ with Lamp-1 (see line 129 and 137).

- 13) Why is Lamp3 so much more promiscuous than Lamp1 and Lamp2 in terms of Ln precipitation? How is this consistent with the proposed mechanism considering that Lamp1 and Lamp3 have the same numbers of D (Lamp3 has in fact one more acidic residue than Lamp 1) and Lamp 2 has a single D and no E? Does poly-D causes Ln precipitation?

First, Supplementary Fig. 10 showing the results of solution turbidity in the original manuscript was deleted because Reviewer 2 pointed out the discrepancy between the solution turbidity and ICP-OES data. Essentially, the analysis of solution turbidity using a spectrophotometer is not always accurate, although it is a readily available technique. The use of this technique for the analysis of mineralization selectivity has the possibility for misinterpretation because it is affected by the reaction time and the size of the generated particles. A re-examination of the results shows that the mineralization selectivity of Lamp-2 against each Ln³⁺ ion is different in the solution turbidity and ICP-OES analyses, whereas Lamp-3 showed almost the same results in both measurements. The ICP-OES results were supported by measurements of the amount of peptide consumed in the same experiment. Therefore, the data concerning the solution turbidity was replaced with accurate results re-measured using ICP-OES (Supplementary Fig. 8). Regarding the first suggestion about the differences in Ln selectivity, we have added a comment in both the Results and Discussion sections (see line 109–113 and 264–270).

Next, regarding the second suggestion about the poly-Asp, as the reviewer pointed out, Asp has an important function for Ln³⁺ recognition. However, Asp itself and LBT3, which contains three Asp (two Glu), failed to cause mineralization. These results indicate that Asp is important for Ln³⁺ recognition, but it alone cannot cause mineralization of Ln hydroxide. Regarding the re-examination using poly-Asp, we are also interested in the effect of poly-Asp for bio-mineralization because previous reports have demonstrated that poly-Asp with MW = 5,000 or 35,400 precipitated struvite (NH₄MgPO₄·6H₂O) and aragonite (CaCO₃) crystals, respectively (References 40 and 51). However, we consider detailed studies using poly-Asp with

various lengths to be beyond the scope of the current study and we have not added data with poly-Asp at this time. We are going to carry out detailed studies of poly-Asp with various lengths, and will report these results in future if it has interesting functions.

- 14) In the same vein, the authors invoke an entropic argument to contrast the behavior of Lamp1 and LBT3 but the difference between the two energies is less than 2 kcal/mol. In addition, Lamp2, which seems to function much like Lamp1 in Ln precipitation has a 2-fold lower entropic component. Some further discussion of these topics would be useful to the reader.

Regarding the first suggestion about the interpretation of the entropy increase, we do not intend to compare the differences in entropy between Lamp-1 and LBT-3, as the thermodynamic data from ITC are sums for the overall process, and there are differences in the reaction mechanisms. The reaction of LBT3 is a simple chelation, and dehydration of water molecules is the reason for the increasing entropy. On the other hand, the reaction of Lamp includes several events: electrostatic interaction, Ln hydroxide generation, complex accumulation, and precipitation. Therefore, the increase in entropy in the Lamp reaction is explained by both the dehydration of coordination water molecules during the electrostatic interaction, and disruption of the hydration shell around the hydrophobic surface during complex accumulation and precipitation. In this revision, we have revised the comments about the ITC experiments in both the Results and Discussion sections (see line 159–177 and 215–227)

Next, regarding the second suggestion about the difference in the thermodynamic parameters for Lamp-1 and -2. As mentioned in response (13), we replaced the solution turbidity data with the results of ICP-OES analysis. As a result, we found that there is a 3–4-fold difference in the reaction stoichiometry of Lamp-1 and the other Lamps. We consider that the difference in reaction stoichiometry should correlate with the difference in the thermodynamic parameters (ΔH) because the magnitude of ΔH should be reflected by the number of Dy bound to Lamp. We have added comments about the thermodynamic parameter and reaction stoichiometry in the Discussion section (see line 261–270).

- 15) It would be interesting for the authors to comment on the economics of the

precipitation process in seawater (1 peptide is consumed to precipitate at best two Ln^{3+}) and on whether or not the Ln concentrations used in the experiments of Fig. 5 (3 mM Dy^{3+}) are realistic. Also, can lanthanides captured on the Lamp1-derivatized sepharose be recovered and the resin regenerated to a functional state?

The Ln captured on the Lamp-1-fused sepharose resin can be easily recovered by treatment with a weak acidic solution (pH 4.0). We had confirmed this recycling process by regenerating the resin 5 times. In this revision, we added this new result about the recycling ability of the sepharose resin (see line 199–201 and Supplementary Fig. 18). However, we have not carried out detailed calculations concerning whether the economy of process opens the possibility for direct recovery from nature without strong acid or alkali. After publishing this manuscript, we plan to work on experiments to confirm the economy of using Lamp-fused materials.

- 16) For how long and at what T was the precipitation experiment of Fig. S22 conducted? Is there a difference at longer time points?

This photograph shows the sample just after mixing the purified proteins (GST-Lamp-1, GST) with Dy^{3+} . There is no difference after at least 1 h of incubation. We have added a comment in this figure legend (see line 232–234 in the Supplementary Information).

- 17) The term aptamers is usually used for nucleic acids not peptides. Typically such peptides are referred to as solid, materials or inorganic binding peptides.

We have revised all instances of ‘aptamer’ as ‘inorganic binding peptide’ or ‘peptide’ according to the reviewer’s instructions.

- 18) The authors should qualify the title of Fig. 4 by calling it a "proposed mechanism"

We have revised the title of Fig. 4 to ‘Proposed mechanism of Ln^{3+} mineralization’ according to the reviewer’s instructions.

Response to Reviewer #2

We thank Reviewer #2 for making some very important suggestions. As the reviewer pointed out, the ITC interpretation is essential for clarification of Ln³⁺ mineralization. According to the Reviewer's comments, we have redone the ITC analysis through trial and error. Based on these results, we have obtained a new and important understanding of the examined system. Our detailed responses are as follows.

- 1) First, the extensive results in the Supplementary Information (22 figures and 5 tables) suggest this is no early breakthrough, but a very well developed project (over a dozen physical methods are used to characterize the lanthanide-peptide complexes), and Nature Communications may not be the most appropriate venue for this work.

As far as we know, this is the first report to demonstrate peptide-based mineralization without artificial reagents for rare earth recovery. Therefore, extensive research is needed to understand this novel phenomenon. Needless to say, lanthanides are important elements for the advanced materials. To deliver this study quickly and openly to researchers over extensive fields, we believe that Nature Communications is the most suitable journal.

Regarding the high number of Supplementary Figures, we have removed three figures and one table from the Supplementary Information through integration or deletion of data, according to the Reviewer's suggestions (19 Figures and 4 Tables).

- 2) Second, in spite of some mechanistic insight (e.g., NMR characterization of the peptide complexes with paramagnetic Dy³⁺ and diamagnetic Ln³⁺), some of the results seem to be rather empirical and some of the models suggest that fundamental coordination chemistry has not been considered for the metal-protein interactions (e.g., the chelate effect is likely to be important here with multiple peptide carboxylates). A number of questions remain about the mechanism for the selective precipitation of lanthanide ions, such as the origin of the lanthanide selectivity of the peptides in Figure S10 and the discrepancy between the data in Figure S9 and S10c.

According to the Reviewer's comments regarding the fundamental coordination chemistry, we revised the discussion section and the references. First, we added a description concerning the interaction between Ln³⁺ and carboxylate of Lamp (see

line 131–133, References 34, 37, and 38). Second, we added a description concerning the coordination of water molecules with Ln^{3+} in the aqueous solution as the origin of the increase in entropy in ITC measurements (see line 159–166, References 34, 37, 42, and 43).

In addition, regarding the second comments about the discrepancy between the data concerning Ln selectivity from the solution turbidity and ICP-OES analyses, we have deleted Supplementary Fig. S10 showing the results of the solution turbidity in the original manuscript because the analysis of the solution turbidity using a spectrophotometer is not always accurate, even though it is a readily available technique. The estimation of the solution turbidity using this method has the potential for misinterpretation because of the influence of the reaction time and the size of the generated particles. A re-examination of the data showed that the mineralization selectivity of Lamp-2 against each Ln^{3+} ion from solution turbidity and ICP-OES analyses were different, whereas Lamp-3 showed almost same results in both measurements. The ICP-OES results were supported by the measurement of the amount of peptide consumed in the same experiment. Therefore, the data concerning the solution turbidity was replaced with an accurate results re-measured using ICP-OES (Supplementary Fig. 8).

- 3) Third, I am concerned about the naive use of isothermal titration calorimetry (ITC) and over interpretation of the ITC data. A fixed stoichiometry of $n = 1$ was used to fit the data for Dy^{3+} binding to Lamp-1 and -2 and Nd^{3+} binding to Lamp-3, when the data in Figure S18 for Dy^{3+} binding to Lamp-1 show the equilibrium ratio to be closer to $n = 2$ (1.62-1.87).

More worrisome, Figure 4 suggests four steps to the formation of lanthanide-peptide precipitate (binding, de-protonation, aggregation, precipitation), and there is no sense of how many of these occur during the ITC measurement. If all of them occur with each injected aliquot, then the experimental thermodynamic data are the sum for the overall process and the interpretation of individual contributions is suspect. For example, what does an experimental K_D really represent if it included the irreversible precipitation of an insoluble species? Comparison of results with Lamp-1 and LBT3 may be instructive, but the difference between -12.7 and -10.9 kcal/mol for the value of $-T\Delta S$ has little molecular significance. It is possible to use ITC to quantify the number of protons lost/gained upon complex formation, and I encourage the authors to exploit this capability for molecule insight on their system.

We understand that the above suggestion is very important for clear elucidation of mineralization by Lamp. According to the reviewer's comments, we have redone the ITC analysis to separate each event. However, it was difficult to separate each event clearly, despite a number of trial and error experiments using two types of ITC (VP-ITC and PEAQ ITC). However, we succeeded in quantifying the number of protons lost/gained upon complex formation by analysing the protonation enthalpy. In this revision, we corrected the title of Figure 4 to 'proposed mechanism' and both the Results and Discussion sections concerning the ITC interpretation were revised drastically to avoid misunderstanding. Individual responses to the reviewer's various points are as follows.

Regarding the first comment about the n value in ITC measurements, there are two reasons to calculate thermodynamic parameters with $n = 1.0$. First, when the titration curve does not show a sigmoidal function, an n value has to be assumed to calculate the thermodynamic parameters according to the manufacturer's protocol (Malvern). Second, Lamp reaction with Dy^{3+} includes several events, which were difficult to separate clearly. Thus, we calculated the thermodynamic parameters using the $n = 1.0$ to simplify the calculation and analysis.

Regarding the second comment about experimental K_D , it would not be appropriate to represent the $1/K$ value calculated from the ITC experiments as K_D because, as the reviewer pointed out, this value is usually used to represent the binding affinity between two molecules. We changed K_D to K to indicate that the reaction constant includes all events occurring during mineralization (see Supplementary Table 4).

Regarding the third comment about the interpretation of the entropy increase, we do not intend to compare the difference in the entropy values of Lamp-1 and LBT-3. As the reviewer pointed out, thermodynamic data obtained from ITC are the sum of the overall process, and there are many differences in the reaction profiles or peptide characteristics of LBT3 and Lamp. The reaction of LBT-3 is a simple chelation, while the reaction of Lamp involves several events. In this revision, we added an extensive explanation about the differences in the reaction profiles and the thermodynamic parameter (ΔH) in the Results section to clarify the interpretation of this data (see line 159–168 and 215–227).

Finally, regarding the fourth comment about quantitation of the number of protons lost/gained upon complex formation, we calculated the protonation enthalpy by ITC analysis referring to previous reports (see line 435–442 and References 45, 46, and 48). In this experiment, we used two kinds of buffer (MES and Bis-Tris), and the results revealed that two protons are released during the reaction (other buffers that are suitable for maintaining pH 6.0, such as phosphate buffer and cacodylate buffer, cause chelation and insoluble precipitates with Dy^{3+}). In this revision, we have added this new data to Supplementary Fig. 15d and revised the Results section concerning the deprotonation reaction (see line 168–177 and 215–227).

- 4) Finally, certain word choice suggests a weak understanding of the molecular processes involved in this phenomenon. For example, the word "rate", which includes a time component (kinetics), is used in several places to describe an equilibrium amount, where the appropriate term would be "extent" or "percentage". The first paragraph of the Discussion has a description that mixes the concepts of kinetics ("slower", "immediately", "rapidly") and thermodynamics ("stability constant", "binding strength"). The authors promote a new term, "interruption of chemical equilibrium", to describe a simple phenomenon whereby La^{3+} coordination by the (chelating) peptide shifts a solution equilibrium to stabilize an insoluble lanthanide-hydroxide complex. I feel that the author's term has a misleading connotation.

We thank Reviewer for this important advice regarding the terminology used in the manuscript. According to the reviewer's suggestions, the word 'rate' in the original manuscript was revised as percentage (see line 108, 193, 689, and 690). Moreover, to avoid confusion, we unified the concept of kinetics and thermodynamics in the first paragraph of the Discussion section to thermodynamics. Additionally, we deleted 'interruption of chemical equilibrium with La^{3+} ', and rewrote the sentence as "stabilization of an insoluble lanthanide hydroxide complex with an artificial peptide" (see line 18, 65, and 279-280).

Response to Reviewer #3

We thank Reviewer #3 for the very enthusiastic remarks. We apologize for our oversight of not including important papers in the discussion of our original manuscript. Several of these studies are cited in the revised manuscript. Our detailed response is as follows.

- 1) The authors introduce lanthanide ion mineralization aptamers (Lamp) for the direct extraction of rare earth ions from solutions by mineralization. The peptides are derived from phage display and disrupt the chemical equilibrium between soluble Ln^{3+} ions and the Ln hydroxide species to afford precipitation, similar to techniques applied for other soluble metal ions and oxides/hydroxides. The literature is appropriately referenced; some reference to simulation studies that characterized the binding mechanisms more specifically are missing and the understanding/comparisons of mechanisms can be refined (e.g. JACS 2012, 134, 6244; Chem. Soc. Rev. 2016, 45, 412).

We wish to express our gratitude for the introduction of these very important papers. We have referred to the simulations of inorganic-bioorganic interfaces in the Discussion section together with reference to the above papers (see line 250-258 and References 55, 56). As the reviewer pointed out, the simulation of molecular recognition is very important for understanding the mechanism of this mineralization phenomenon. In fact, we have started molecular simulations, but we have not obtained clear results at this time because several parameters are insufficient for lanthanide elements. In the near future, we plan to try to improve Lamp through molecular simulations of inorganic-bioorganic interfaces with reference to the above studies.

- 2) The role of Cys in the chosen peptides is not clear - Cys is somewhat attracted to metal, although mostly to elemental metals that are not present here (see e.g. Soft Matter 2011, 7, 2113; JACS 2013, 135, 11048). The biopanning approach itself appears to be carried out with care and lead to strongly binding sequences. It may be hard to say that these would be the strongest possible binding sequences, however, as the phage library for a 12 peptide covers only 10^9 out of $20^{12} \sim 10^{16}$ peptides (this is less than one-millionth of possibilities). In fact, the reported peptides are even up to 16 amino acids long (Lamp no 3, p. 4). A comment on the limitations would be suitable here and not at all affect the credibility or impact of the manuscript.

In aqueous solution, it has been reported that the cyclic structure of peptides formed by an intra-disulfide bond with two Cys residues decreases the conformational entropy compared with that of a linear structure. The decrease in conformational entropy is expected to result in higher binding ability for the target (References 28–30). Therefore, we designed a cyclic peptide library to select for higher binding ability to achieve mineralization of rare earth ions. In this revision, we have added an explanation of why we adopted the cyclic peptide (see line 74–77).

As Reviewer pointed out, bio-panning is a valuable approach for selecting optimum ligands. However, we do not think that the three kinds of Lamp in this study are the most useful peptides for recovery of the rare earth elements because, although our concept was confirmed, our selection was performed using a peptide library with limited conformational diversity. We expect that more efficient peptides can be obtained based on the findings that have been obtained with Lamp. Currently, we are trying to achieve improvement of the peptide using a mutated Lamp library (see line 281–284).

- 3) **Typographical: line 122: "chemical shifts" into "chemical shifts"**

We apologize for our carelessness. We have corrected ‘chemical shifts’ as ‘chemical shifts’.

- 4) **The interpretations of the binding mechanism on the basis of NMR chemical shifts are carefully performed and helpful to ascertain dominant interactions. Isothermal titration adds valuable information as to the pH dependence of the mineralization point and the general prevalence of protonated/deprotonated species.**

We thank the reviewer for the valuable comments regarding our study.

- 5) **On p. 8 it may be noted that the Lamp peptides are similar to oxide recognizing/forming peptides such as silica or titania binders. They have virtually nothing in common with metal binding peptides as there is no elemental metal in the process here. The authors might want to clarify that metal-binding peptides recognize metal surfaces by soft epitaxy, not by ion pairing or electrostatic interactions (see Chem. Soc. Rev. 2016, 45, 412; or original studies such as the ref.**

in *Soft Matter* 2011 above, *Nano Lett.* 2013, 13, 840; *Adv. Funct. Mater.* 2015, 25, 1374).

Lamp resembles silica and titania binding peptides with respect to the generation of amorphous precipitates (References 55–58). In addition, electrostatic interactions are also important for the function of Lamp and these peptides rather than soft epitaxy. Therefore, the mineralization mechanism seems to be partially similar. However, Lamp is obviously different from these peptides in two respects. (1) The pI of Lamp is a 3.3–3.8, while that of the silica binding peptide is 8.6–9.6, and that of the titania binding peptide is 6.2–12.4. (2) Lamp recognizes a hydrated ion, while the silica or titania binding peptides recognize ions coordinated by organic components. In this revision, we have added new comments concerning this recognition in the Discussion section together with important references (see line 250–259 and References 55–58).

- 6) p. 7-9: The net entropy increase upon peptide binding to oxide/hydroxide nuclei due to freed water molecules is consistent with earlier studies of peptide binding to silica and metal surfaces (*JACS* 2009, 131, 9704; *Chem. Mater.* 2014, 26, 5725).

As the reviewer pointed out, the net entropy increase in the mineralization reaction is an important point for our system, which is consistent with metal-binding peptide as mentioned above. Therefore, we have not commented on the net entropy increase for peptide binding to metal surfaces to avoid misunderstanding, but we have added this paper as an important reference for metal recognition with soft epitaxy (References 54 and 55).

- 7) p. 9-11: Again, I would like to remind the authors not to confuse metal nanocrystal growth with oxide growth. The acting peptide recognition, surface chemistry, and growth mechanisms are fundamentally different (soft epitaxy independent of pH versus ionic and pH sensitive acid-base chemistry). To explain the extraction of metal ions with the chosen aptamers, the recognition of aptamers on elemental metal substrates is only marginally relevant (see comment 5 above).

We thank reviewer for this valuable advice. In this revised manuscript, we have taken care to indicate that the mineralization event with Lamp is not metal nanocrystal growth but oxide growth (see line 246–249). Additionally, we believe

that this difference is a unique phenomenon in this study.

- 8) Comments on methods and characterization: This manuscript is a very accurately informed account of the stability of metal ions, hydroxides, and mineralization of lanthanide metal ions with peptides. Surface characterization of Dy₂O₃ and Nd₂O₃ nanoparticles by FTIR shows the presence of OH surface termination in solution. Biopanning and turbidity measurements are well documented. TEM, SEM, and XAFS data characterize the elemental composition of precipitates; the pH sensitivity was tested; NMR chemical shifts were measured and meticulously assigned, including 2D TOCSY spectra and clear identification of peptide residues in contact with metal ions (such as for Dy³⁺). Binding constants are calculated including clear designation of error bars. Thermodynamic analyses of the reactions are included. EDX was employed to verify the accumulation of metal ions in the precipitates. Mineralizing protein aptamers and originating phage DNA were analyzed. Binding strengths and error bars of peptides to Ln³⁺ and hydroxylated Ln₂O₃ are reported in exceptional precision - the study appears by far more extensive and meticulously performed than many others reported previously for other substrates.

We thank the reviewer for these valuable comments regarding our study.

- 9) The TOC graphic is appropriate. The movie is somewhat plain and simple, but conveys the message.

Summary: Scientifically, this paper would in my opinion rank in the top 5%. The relevance of the lanthanides may be somewhat debated, although I concur with the authors that the specialty elements already find many niche applications and will remain in demand for the foreseeable future. Given the high cost, recovery by mineralization, or at least knowing possible binding constants and pathways to mineralize dissolved ions are important knowledge, and this manuscript clearly advances the field by providing example protocols and valuable reference data. I recommend the authors to perform suggested (minor) revisions and address all mentioned concerns.

We thank the reviewer for these valuable comments regarding our study. In the near future, we plan to try improving Lamp through molecular simulations of inorganic-bioorganic interfaces.

Reviewers' comments:

Reviewer #1 (Remarks to the Author):

The revised manuscript adequately addresses my concerns and I am happy to endorse publication.

Reviewer #2 (Remarks to the Author):

In this revised version of the manuscript, the authors have made a good effort to address the concerns that I and the other reviewers had about this study. It is clear, however, that this team needs an inorganic coordination chemist to help with the explanations and models for these experimental phenomena. For example, while the entropic contributions to lanthanide binding from dehydration of the metal ion (and peptide) are included in the discussion/explanation, the entropic penalty from conformationally constraining the Lamp peptide upon lanthanide coordination by two or more carboxylates is not.

With regard to the calorimetric (ITC) measurements, I am pleased to see that additional measurements have been made in another buffer and analyzed to reveal that ~ 2 protons are released/displaced upon Dy(III) binding to Lamp-1, as this further supports and quantifies hydroxo formation upon binding. However, armed with this new information, the thermodynamic values in Table S4 have not been adjusted to subtract buffer (MES) protonation from the experimental enthalpy (see J. Biol. Inorg. Chem. 2010 15, 1183-1191). So, the ΔH and $-\Delta S$ values in this table are still only experimental values appropriate for pH 6.1 and 50 mM MES buffer. Subtraction of the contribution from buffer protonation, $1.9 \times (\sim 3.5 \text{ kcal/mol}) = \sim 7 \text{ kcal/mol}$, would provide essentially buffer independent thermodynamics at this pH. (Please check the K value for Dy(III) and LBT3, as this did not change from the previous version while the other K values did.)

Figure 4 shows a proposed multi-step mechanism for formation of the Ln(III)-hydroxo-Lamp precipitate. An important question is how this relates to the calorimetric data (Fig 3b and Fig S15), since the mechanism has an irreversible final step. Is there any visible precipitate or any DLS (dynamic light scattering) or TEM evidence of aggregation in samples removed from the calorimeter after ITC measurements? If not, then aggregation and precipitation are slower than binding and deprotonation, and the ITC data indicate the equilibrium formation of soluble Ln(III)-hydroxo-Lamp species (first few steps in Fig. 4).

Finally, in the expanded caption to Figure 2 it is noted that samples used for the data in Fig. 2f contain 100 mM MES buffer. Why is a buffer present when the goal is to determine the change in pH upon mixing Dy(III) and Lamp-1?

Reviewer #3 (Remarks to the Author):

In response to comment 2, the authors still need to acknowledge the challenges of combinatorial screening. 16-mer peptides allow $6.5 \cdot 10^{20}$ ($=20^{16}$) different peptide sequences, of which only about 109 are covered in the phage library and thus in this study. This is clearly a major systematic limitation, and even though it is hard to find better methods, this limitation must be mentioned. It only makes the paper stronger and helps the community to think of what is currently being missed and how to improve it.

If the authors continue to refuse clearly laying out this limitation in the main text of the manuscript in the next round of revision, this work should be rejected. It is Nature policy to disclose limitations of

methods.

In response to comment 5, the authors have begun to clarify that Lamp peptides – as an oxide binder – have very little in common with peptides binding to elemental metals. As also other referees suggested, the manuscript is weak on explaining responsible interactions for which general concepts are well known (ref. 55). More detail on the molecular mechanisms, or likely molecular mechanisms, of binding should be given backed up by literature and own observations. For certain, metal surface chemistry is qualitatively so different from that of oxides and known to result in completely different peptide-surface interactions; so at least this can be made clear to avoid obviously misleading statements.

I think that, upon addressing further comments from all referees as requested, this manuscript is publishable and could be a meritorious contribution for Nature Communications.

Response to Reviewer #2

We thank the Reviewer #2 for these very important suggestions. As the reviewer pointed out, inorganic coordination chemistry is essential for clarifying the Ln^{3+} mineralization process. According to the reviewer's suggestion, we have reconsidered the mineralization mechanism using ITC (isothermal titration calorimetry) data. In this revision, we have added new sentences, which are highlighted in yellow in the text file. Our detailed responses are as follows.

- 1) In this revised version of the manuscript, the authors have made a good effort to address the concerns that I and the other reviewers had about this study. It is clear, however, that this team needs an inorganic coordination chemist to help with the explanations and models for these experimental phenomena. For example, while the entropic contributions to lanthanide binding from dehydration of the metal ion (and peptide) are included in the discussion/explanation, the entropic penalty from conformationally constraining the Lamp peptide upon lanthanide coordination by two or more carboxylates is not.

According to the reviewer's suggestion, we asked a coordination chemistry professional in our organization for advice on the mineralization mechanism of Lamp. Referring to this discussion, we have revised the text discussing the coordinated water molecule and aggregation through hydrophobic effect.

In this phenomenon, the reaction of Lamp and Ln^{3+} is enthalpically unfavourable ($\Delta H_{\text{total}} > 0$) because the enthalpic penalty for removing the coordinated water molecules from Ln^{3+} ($\Delta H_{\text{water}} > 0$) is greater than the energy of ionic bond formation between Lamp and Ln^{3+} , which is intrinsically enthalpically favourable ($\Delta H_{\text{ion}} < 0$). On the other hand, the release of water molecules (dehydration) is entropically favourable ($-T\Delta S_{\text{water}} < 0$). Additionally, the disintegration of the hydration structure by aggregation/accumulation owing to the hydrophobic effect of Lamp is also entropically favourable ($-T\Delta S_{\text{aggre}} < 0$). These reactions over compensate for the conformational entropy loss of the peptide with Ln^{3+} recognition ($-T\Delta S_{\text{conf}} > 0$, $-T\Delta S_{\text{total}} = -T(\Delta S_{\text{water}} + \Delta S_{\text{conf}} + \Delta S_{\text{aggre}}) < 0$). This entropic advantage derived from water release is greater than the above enthalpic penalty, resulting in a spontaneous reaction.

$$\begin{aligned}\Delta G &= \Delta H_{\text{total}} - T\Delta S_{\text{total}} \\ &= (\Delta H_{\text{ion}} + \Delta H_{\text{water}}) - T(\Delta S_{\text{water}} + \Delta S_{\text{conf}} + \Delta S_{\text{aggre}}) < 0\end{aligned}$$

ΔH_{ion} : enthalpy change for binding between Lamp and Ln^{3+}

ΔH_{water} : enthalpy change for dehydration

ΔS_{water} : entropy change for dehydration

ΔS_{conf} : entropy change for the peptide conformation

ΔS_{aggre} : entropy change for aggregation (particle generation)

T : absolute temperature

According to the reviewer's comment about the entropic penalty, we have added text according to the above explanation in the discussion section of the revised manuscript (see lines 229–234).

- 2) With regard to the calorimetric (ITC) measurements, I am pleased to see that additional measurements have been made in another buffer and analyzed to reveal that ~2 protons are released/displaced upon Dy(III) binding to Lamp-1, as this further supports and quantifies hydroxo formation upon binding. However, armed with this new information, the thermodynamic values in Table S4 have not been adjusted to subtract buffer (MES) protonation from the experimental enthalpy (see J. Biol. Inorg. Chem. 2010 15, 1183-1191). So, the ΔH and $-T\Delta S$ values in this table are still only experimental values appropriate for pH 6.1 and 50 mM MES buffer. Subtraction of the contribution from buffer protonation, $1.9 \times (\sim 3.5 \text{ kcal/mol}) = \sim 7 \text{ kcal/mol}$, would provide essentially buffer independent thermodynamics at this pH. (Please check the K value for Dy(III) and LBT3, as this did not change from the previous version while the other K values did.)

According to the reviewer's comments, we have added the ITC data using another buffer, Bis-Tris, to Supplementary Figure 16. Additionally, we have removed the data for the protonation enthalpy from the same figure (see Supplementary Figure 16) and added the control data (only buffer) and the suggested reference (see Supplementary Figure 15d and Reference 49).

Regarding Supplementary Table 4, we have added the new thermodynamic parameters, from which the buffer (MES) protonation effect has been subtracted (see Supplementary Table 4, Lamp-1^c). Additionally, we have reanalysed all of the ITC data through curve fitting using the VP-ITC software (Origin 7.0). As a result, the ΔH value has changed slightly, and we have corrected the sentences referring to the ΔH value in the results section of the revised manuscript (see lines 181 and 183). In the first manuscript (submitted 2016/5/6), the K value for the reaction of Dy(III) and LBT3 was incorrect. Therefore, we have already revised the K value of 477×10^3 as 477×10^4 in the second manuscript (resubmitted 2016/8/26). Moreover, the units of all the K values were adjusted to 10^4 M^{-1} .

- 3) Figure 4 shows a proposed multi-step mechanism for formation of the Ln(III)-hydroxo-Lamp precipitate. An important question is how this relates to the calorimetric data (Fig 3b and Fig S15), since the mechanism has an irreversible final step. Is there any visible precipitate or any DLS (dynamic light scattering) or TEM evidence of aggregation in samples removed from the calorimeter after ITC measurements? If not, then aggregation and precipitation are slower than binding and deprotonation, and the ITC data indicate the equilibrium formation of soluble Ln(III)-hydroxo-Lamp species (first few steps in Fig. 4).

After the ITC measurements, a visible precipitate was observed in the solution that contained 90 μM of Lamp-1 and 2 mM of Dy^{3+} . The generation of visible particles at this concentration correlates with the results shown in Supplementary Figure 4. To avoid any misunderstandings, we have added a sentence about the formation of a precipitate in the result sections and the figure legend (see lines 175 and 694–696). Regarding evidence for aggregation, we attempted DLS (dynamic light scattering) measurements using a Zetasizer Nano ZSP (Malvern) instrument according to the reviewer's suggestion. However, the results were difficult to analyse because of the detection limit of this device at the concentrations used for the ITC measurements.

- 4) Finally, in the expanded caption to Figure 2 it is noted that samples used for the data in Fig. 2f contain 100 mM MES buffer. Why is a buffer present when the goal is to determine the change in pH upon mixing Dy(III) and Lamp-1?

This experiment was carried out using 0.1 mM MES buffer, not 100 mM (see line 684). We used 0.1 mM MES buffer in this experiment because it is too difficult to adjust the initial pH without buffer.

Response to Reviewer #3

We wish to express our gratitude for the reviewer's comments on some serious points. We apologize that our response to the reviewer's suggestions was not sufficient in the previous revision. In this revision, we have made careful corrections in the manuscript, which are highlighted in yellow. Our detailed responses are as follows.

- 1) In response to comment 2, the authors still need to acknowledge the challenges of combinatorial screening. 16-mer peptides allow $6.5 \cdot 10^{20}$ ($=20^{16}$) different peptide sequences, of which only about 10^9 are covered in the phage library and thus in this study. This is clearly a major systematic limitation, and even though it is hard to find better methods, this limitation must be mentioned. It only makes the paper stronger and helps the community to think of what is currently being missed and how to improve it. If the authors continue to refuse clearly laying out this limitation in the main text of the manuscript in the next round of revision, this work should be rejected. It is Nature policy to disclose limitations of methods.

As the reviewer pointed out, in the case of 16-mer peptides, the theoretical diversity of the peptide library is 6.5536×10^{20} ($=20^{16}$). However, in our study, the four amino acids at both ends of the peptide libraries are fixed (SCXXX---XXXCS: the number of X is 9–12; two Cys residues cause the formation of the intra-disulfide bond). Hence, the theoretical diversity of these peptide libraries is 5.12×10^{11} – 4.096×10^{15} (20^9 – 20^{12}). We of course understand that the diversity of our peptide libraries in this work was still low (1.56×10^6 – 3.76×10^7), similar to most previous studies on the isolation of metal binding peptide using peptide libraries, where the diversity was approximately 1.0×10^7 – 1.0×10^9 (references 18, 21, and 22). In this revision, we have added a description about the systematic limitation of our peptide library according to the reviewer's comment (see lines 84-86 and 299).

[UNPUBLISHED DATA REDACTED FROM PEER REVIEW FILE BY EDITORIAL TEAM UPON AUTHOR REQUEST]

Of course, the consensus motif of Lamp-1 is not understood yet (see lines 92-95), and its clarification requires a more detailed examination. Therefore, we cannot describe about the consensus motif of Lamp-1 at this time. However, even if the diversity has systematic limitations, it is theoretically possible to screen target specific peptides with an efficient selection technique.

Regarding the reviewer's important comments about why we were able to isolate Lamp in spite of these systematic limitations, we consider the following three points to be important for the efficient isolation of the target-specific peptide.

- 1) We selected the T7 phage display system, which displays an average of 5–15 copies of the peptides on the phage surface. This multiple display system enables the isolation of a peptide binder, even when the affinity is not high because of rebinding and the avidity effect.
- 2) A cyclic peptide platform reduces unfavourable changes in conformational entropy. This function provides steady binding for the target molecule.
- 3) Considering previous studies, in which inorganic binding peptides tend to contain more hydrophilic amino acids than hydrophobic amino acids, we constructed a peptide library with the NNK codon, which has a tendency to include 56.25% hydrophilic amino acids.

We apologise for our oversight in the previous manuscript of not including a description of the above points. According to the reviewer's comments, we have added a detailed description in the results and methods sections, as well as new references (see lines 75–88, Supplementary information lines 261–263, and reference 33 and 34).

- 2) In response to comment 5, the authors have begun to clarify that Lamp peptides – as an oxide binder – have very little in common with peptides binding to elemental metals. As also other referees suggested, the manuscript is weak on explaining responsible interactions for which general concepts are well known (ref. 55). More detail on the molecular mechanisms, or likely molecular mechanisms, of binding should be given backed up by literature and own observations. For certain, metal surface chemistry is qualitatively so different from that of oxides and known to result in completely different peptide-surface interactions; so at least this can be made clear to avoid obviously misleading statements.

As the reviewer suggested, we also consider that the mechanism of Lamp binding to the Ln hydroxide surface is important for the subsequent mineralization reaction. To clarify the binding kinetics, we attempted QCM measurements (quartz crystal

microbalance; Initium) to analyse the interaction between Lamp and Ln hydroxide of the surface of the Ln oxide nanoparticles. However, effective data were not obtained at this time because the immobilized nanoparticles were gradually removed during the measurement.

It has been reported that the surface of Ln oxide nanoparticles is hydroxylated easily, although the degree of hydroxylation depends on the particle size (reference 35). In this study, we also confirmed that the Ln oxide nanoparticles, which were used as target for peptide selection, have hydroxylated surfaces (Supplementary Figure 2). As the result of using these materials for selection, we succeeded in isolating the peptides that recognize Ln hydroxide. In fact, ELISA (enzyme-linked immunosorbent assay) measurements showed that Lamp strongly recognizes hydroxylated Ln oxide nanoparticles, unlike the other control peptides. For example, the binding ability of Lamp-1 to Ln hydroxide was approximately 242-fold stronger than that of RE-1, an Ln oxide binding peptide (reference 37) (Supplementary Table 3). To improve clarity, we have added some text about “Ln hydroxide” recognition in the results section (see lines 88–92, 96 and 223–224, and Supplementary information lines 38–39).

Regarding the molecular mechanism of Lamp-based mineralization, we received a similar suggestion from Reviewer #2. According to the reviewer’s suggestion, we asked a surface chemistry professional in our organization for advice about Lamp recognition. Referring to this discussion, we reconsidered the discussion regarding the dehydration of coordinated water molecules by electrostatic interactions and the disintegration of the hydration structure by aggregation/accumulation through the hydrophobic effect. As far as we know, this is the first report to demonstrate peptide-based mineralization under mild conditions without artificial reagents for rare earth recovery. The essence of this reaction is the stabilization of an insoluble lanthanide hydroxide complex by electrostatic interactions, which is different from crystalline growth of oxide nanoparticles. In this revision, we revised our considerations about the conformational entropy change of Lamp and the enthalpy change with Ln³⁺ recognition in the discussion section (see lines 229–234, 254–259, and 264–274).

Reviewers' comments:

Reviewer #2 (Remarks to the Author):

The authors have adequately addressed all of my concerns in this latest version of their manuscript. This is an interesting and well supported study that provides the basis for some potentially useful applications.

Reviewer #3 (Remarks to the Author):

After the 2nd round of review, the authors are still reluctant to improve the manuscript as requested.

With regard to the comment on limited coverage of the peptide library (<0.01% of possible peptide sequences), still only two lines were added in the manuscript while the authors are trying to diffuse the comment in 2 long pages of reply that will never be read by the audience. While I would not consider the low coverage of theoretically possible peptide sequences a detrimental problem, I do consider it a problem that the authors are unwilling to discuss extremely obvious limitations. Even the arguments presented are not convincing; it is quite possible that only a millionth or a billionth of possible peptides are covered in this study.

Then, the comment on binding mechanisms in addressed equally poorly. Here the authors refuse for the 2nd time to acknowledge prior work on peptide binding mechanisms on similar oxide surfaces such as silica and titania (for which I also gave references). The surface chemistry of lanthanide oxides is similar, there are M-OH groups, there are MO⁻ (alkali⁺) groups, and the mechanisms are known: ion pairing at higher pH, hydrogen binding, and less specific hydrophobic interactions at lower pH. The critical constant is the pK value of the M-OH groups, or the protonated M-OH₂⁽⁺⁾ groups, respectively. Accordingly one can design peptides, at least provide some guidance.

It is very interesting to hear that the authors consulted an "expert" to look into this matter while these facts are already published. It is unclear why such related information is not cited, either. It is correct that no one may have looked at lanthanide oxide/hydroxide surfaces in particular, however, the mechanisms are not going to change - just as they are the same for silica (main group) and titania oxide and hydroxide surfaces (transition metal).

I am very surprised at the unwillingness of the authors to accept the current state of the art, or literally any well-intended suggestions to improve the manuscript.

Response to Reviewer #3

We wish to express our gratitude to the reviewer for comments on some important points, which have helped us to significantly improve our manuscript. We apologize that our response to the reviewer's suggestions was not sufficient in the previous revision. In this revision, we have made careful corrections regarding the systematic limitations and the metal surface, which are highlighted in yellow.

1) After the 2nd round of review, the authors are still reluctant to improve the manuscript as requested.

With regard to the comment on limited coverage of the peptide library (<0.01% of possible peptide sequences), still only two lines were added in the manuscript while the authors are trying to diffuse the comment in 2 long pages of reply that will never be read by the audience. While I would not consider the low coverage of theoretically possible peptide sequences a detrimental problem, I do consider it a problem that the authors are unwilling to discuss extremely obvious limitations. Even the arguments presented are not convincing; it is quite possible that only a millionth or a billionth of possible peptides are covered in this study.

In this study, we have no intention to dissemble our data and ideas. According to the important suggestion about the limitation of peptide screening, we have added comments about the following points in the discussion section.

(1) The peptide library used in this study has limited structural diversity (see lines 222–224).

(2) The limited diversity likely results in a low number of isolated peptides (see lines 223).

(3) The screening strategies allow for effective peptide isolation, even in such a limited system (see lines 224–227).

(4) The peptides presented in this study are not the best sequences, and leave room for optimization (see lines 313–317).

We believe that our opinions listed above are sufficient to allow the audience to understand the limitations of peptide screening using a phage display system, as well as the effective approach for peptide isolation under such conditions.

2) Then, the comment on binding mechanisms is addressed equally poorly. Here the authors refuse for the 2nd time to acknowledge prior work on peptide binding mechanisms on similar oxide surfaces such as silica and titania (for which I also gave references). The surface chemistry of lanthanide oxides is similar, there are M-OH groups, there are MO- (alkali+) groups, and the mechanisms are known: ion pairing at higher pH, hydrogen binding, and less specific hydrophobic interactions at lower pH. The critical constant is the pK value of the M-OH groups, or the protonated M-OH₂(+) groups, respectively. Accordingly one can design peptides, at least provide some guidance.

It is very interesting to hear that the authors consulted an "expert" to look into this matter while these facts are already published. It is unclear why such related information is not cited, either. It is correct that no one may have looked at lanthanide oxide/hydroxide surfaces in particular, however, the mechanisms are not going to change - just as they are the same for silica (main group) and titania oxide and hydroxide surfaces (transition metal).

We thank the reviewer for the very important suggestion about metal surface chemistry. According to the reviewer's recommendation, we read the previous papers again, especially *JACS*, **136**, 6244-6256 (2012) and *Chem. Mater.* **26**, 5725-5734 (2014).

It has previously been reported that Ln-oxide and Ln-hydroxide surfaces have positive charges in solution at pH 7.5 (new ref. 51). In our experimental condition, this means that Ln-OH and the protonated LnOH₂(+) are distributed on the (hydro) Ln-oxide surface. Therefore, we have newly discussed that the Lamp recognizes (hydro) Ln-oxide mainly by ion pairing and hydrogen bond formation, as suggested by the reviewer. In this revision, in the discussion section, we have added a description of the peptide binding mechanism on the (hydro) Ln-oxide surface with reference to previous studies and known mechanisms for silica and titania binding peptides (see lines 227-236 and 275-290, and references 51, 60, and 61).

REVIEWERS' COMMENTS:

Reviewer #3 (Remarks to the Author):

The authors have now addressed the previous comments and I think also recognized the value of known adsorption mechanisms of peptides on oxide and metal substrates. For example, the role of $\text{Ln}(\text{OH})_2^+$ and $\text{Ln}(\text{OH})$ groups is now recognized and points readers in a direction of rational understanding. The new explanation of peptide binding via ion pairing and hydrogen bonding on the lanthanide oxides/hydroxides versus soft epitaxy on elemental metals is also found to be consistent with the microscopy and sequence data presented, and updates to the current state of knowledge/makes this manuscript stronger.

I also appreciate the authors mentioning more clearly the limitations in the number of peptides covered by current combinatorial libraries, which is not a small limitation.

Demonstrating the current state of the art and pointing out its limitations, in my view, is indeed a strength rather than a weakness. Also, the body of novelty in this work is already impressive. Being frank about the limitations in no way limits the significance of the study, in contrast, it also allows the authors and other researchers to think about possible follow-ups and developing even more powerful techniques.

In conclusion, I would like to congratulate the authors for this novel, meticulous, and informative study of the Lamp peptides and rare earths mineralization. I believe this work can be a landmark and first contribution in this area with potentially great impact.